# Plasma p-tau212 antemortem diagnostic performance and prediction of autopsy verification of Alzheimer's disease neuropathology

Przemysław R. Kac [1] ✉, Fernando González-Ortiz [1,2], Andreja Emeršič[3,4], Maciej Dulewicz [1], Srinivas Koutarapu[1], Michael Turton[5], Yang An[6], Denis Smirnov[7], Agnieszka Kulczyńska-Przybik[8], Vijay R. Varma [9], Nicholas J. Ashton[1,10,11,12], Laia Montoliu-Gaya[1], Elena Camporesi[1], Izabela Winkel[13], Bogusław Paradowski[14], Abhay Moghekar[15], Juan C. Troncoso[15,16], Tammaryn Lashley [17], Gunnar Brinkmalm [1], Susan M. Resnick [18], Barbara Mroczko[8], Hlin Kvartsberg [1,2], Milica Gregorič Kramberger[3,19,20], Jörg Hanrieder [1,21], Saša Čučnik[3,4,22], Peter Harrison[5], Henrik Zetterberg [1,2,21,23,24,25], Piotr Lewczuk[8,26], Madhav Thambisetty [9], Uroš Rot[3,19], Douglas Galasko[7], Kaj Blennow [1,2,28] & Thomas K. Karikari [1,27,28]

Blood phosphorylated tau (p-tau) biomarkers, including p-tau217, show high associations with Alzheimer's disease (AD) neuropathologic change and clinical stage. Certain plasma p-tau217 assays recognize tau forms phosphorylated additionally at threonine-212, but the contribution of p-tau212 alone to AD is unknown. We developed a blood-based immunoassay that is specific to p-tau212 without cross-reactivity to p-tau217. Here, we examined the diagnostic utility of plasma p-tau212. In five cohorts (*n* = 388 participants), plasma p-tau212 showed high performances for AD diagnosis and for the detection of both amyloid and tau pathology, including at autopsy as well as in memory clinic populations. The diagnostic accuracy and fold changes of plasma p-tau212 were similar to those for p-tau217 but higher than p-tau181 and p-tau231. Immunofluorescent staining of brain tissue slices showed prominent p-tau212 reactivity in neurofibrillary tangles that co-localized with p-tau217 and p-tau202/205. These findings support plasma p-tau212 as a peripherally accessible biomarker of AD pathophysiology.

The recent development of blood biomarkers that target principal pathological hallmarks of Alzheimer's disease (AD), including amyloid beta (Aβ) and phosphorylated tau (p-tau), have offered diagnostic and prognostic opportunities that were not feasible using cerebrospinal fluid (CSF) or neuroimaging biomarkers[1–5]. In blood, however, plasma p-tau is the leading candidate for such evaluation. Plasma Aβ biomarkers have high diagnostic accuracies but modest changes in Aβ-positive compared with Aβ-negative individuals, leading to poor clinical robustness[6,7]. In contrast, recent findings from multiple independent cohorts have shown that plasma p-tau

biomarkers (p-tau181, p-tau217, and p-tau231) are strongly associated with both brain Aβ load[8–10], the amounts and severity of tau pathology across the AD continuum[8], and predict longitudinal changes in brain Aβ pathology[11]. Importantly, in contrast to plasma Aβ, high-performing p-tau immunoassays have a larger fold change (>100%) between Aβ-positive compared with Aβ-negative individuals, especially in the advanced symptomatic stages[12]. Moreover, the plasma p-tau forms accurately identify an abnormal Aβ-PET scan[13], a positive neuropathological diagnosis of AD at autopsy[14,15], and predict longitudinal cognitive change[13]. Furthermore, the plasma p-tau variants have shown promise as first-line screening tools to identify community-dwelling older adults at risk of future cognitive impairment and/or AD pathology at the group level[16,17]. Given these high performances[15], plasma p-tau biomarkers are being employed as surrogate markers of brain Aβ and tau pathology in the recruitment and monitoring of participants during anti-amyloid therapeutic trials[18]. Anti-amyloid clinical trial participants who showed time- and dose-dependent reductions in brain Aβ levels also had a concurrent mean decrease in plasma p-tau of more than 20%[18]. These high diagnostic and prognostic accuracies, combined with strong analytical robustness[7,11], make plasma p-tau highly attractive for clinical use and will have important roles to play in the era of anti-AD therapies, where clinicians will need to confirm the presence of the pathology before cognitively impaired patients can be prescribed Food and Drug Administration (FDA)-approved anti-amyloid drugs[19].

In head-to-head comparison studies, plasma p-tau217 has often demonstrated higher accuracies for both Aβ and tau pathology than p-tau181 and p-tau231[15] both for baseline and longitudinal applications, especially when focusing on cognitively unimpaired participants some of whom might have subthreshold deposition of Aβ aggregates in defined brain regions characteristic of AD[11,12]. In contrast, two recent studies showed that p-tau231 is significantly increased at an earlier stage of AD with lower amounts of amyloid pathology as compared with p-tau217 and p-tau181[11,13]. Antemortem plasma p-tau217 had stronger associations with the amounts of Aβ plaques and tau tangles at autopsy, compared with p-tau181 and p-tau231[15]. One study suggested that plasma p-tau217 – but not p-tau181 or p-tau231 – had equivalent accuracies as CSF p-tau217 to identify participants with increased levels of brain Aβ and tau aggregates assessed by positron emission tomography (PET)[20]. A recent study reported that plasma p-tau217 showed the best performances to track longitudinal changes in brain Aβ levels, despite the levels of other p-tau forms also increasing according to Aβ-PET uptake at baseline; thus, p-tau217 could be advantageous as a positive outcome predictor for anti-Aβ therapies[11]. Yet, several studies have reported that plasma p-tau217 has poor analytical sensitivity in groups with low levels of these biomarkers (i.e., cognitively unimpaired) with up to 50% samples below the LLOQ[21,22]. Further investigation is needed to understand if other p-tau epitopes may have similarly high – or even superior – performances to p-tau217.

We hypothesized that phosphorylation at other unstudied epitopes may provide additional information on AD pathophysiology and neuropathology. To this end, computational prediction studies identified threonine-212 as a leading target in the tau molecule for kinases[23] (Supplementary Fig. 1), a finding that is supported by multiple publications[24–26]. Notably, some kinases did phosphorylate tau at threonine-212, but not at threonine-217[27,28]. On the other hand, different kinases phosphorylate tau at both threonine-212 and threonine-217[29], further suggesting shared molecular properties. The main aim of this study was, therefore, to investigate if tau phosphorylation at threonine-212 is an indicator of AD neuropathology, to develop and validate a blood biomarker to quantify these pathological changes, and to compare its performances with that of plasma p-tau217. To achieve this, we generated two sheep monoclonal antibodies (mAbs) that specifically recognize p-tau217 or p-tau212 and do not require protein phosphorylated at both positions.

In this work, we evaluated the immunohistochemical staining properties of p-tau212 versus p-tau217 against neurofibrillary tangles in autopsy-verified AD brain tissues. Next, we developed a plasma assay for p-tau212 and examined its diagnostic accuracies in five independent cohorts relative to plasma p-tau217, including those with autopsy verification as well as others from real-world memory clinic settings.

## Results

### The p-tau212 and p-tau217 antibodies are highly specific to phosphorylation at the indicated sites

We developed two sheep monoclonal antibodies that are selective for p-tau212 and p-tau217 independently. Following successful production and purification, the generated antibodies were screened against synthetic tau peptides or recombinant tau proteins that were either non-phosphorylated or phosphorylated at given epitopes. In further experiments, we titrated the antibody amounts against an identical concentration (500 ng/mL) of each synthetic peptide or recombinant protein. The p-tau212-specific antibody did selectively bind peptides that were phosphorylated at threonine-212, irrespective of the neighboring phosphorylated sites (Fig. 1). This includes peptides phosphorylated only at threonine-212 (Fig. 1a), both at threonine-212 and threonine-217 (Fig. 1c), and additionally at other sites in recombinant full-length tau-441 in vitro phosphorylated by dual specificity tyrosine phosphorylation regulated kinase 1 A (DYRK1A; Fig. 1e). However, the antibody did not recognize peptides that were phosphorylated exclusively at threonine-217 (Fig. 1b) or those not phosphorylated at any site in full-length tau-441 (Fig. 1f), demonstrating specificity to threonine-212. Similarly, the p-tau217-specific antibody, only recognized p-tau217-positive peptides and proteins (Fig. 1b, c, e) but not those either phosphorylated only at threonine-212 (Fig. 1a) or not phosphorylated at all (Fig. 1f). Both antibodies recognized p-tau212+p-tau217-positive constructs (Fig. 1c) but showed much reduced affinity to a peptide that was additionally phosphorylated at serine-214, as this may hinder binding access (Fig. 1d). Moreover, the p-tau212 and p-tau217 antibodies did not bind to peptides phosphorylated at the epitopes serine-214, threonine-181 and threonine-231 (Fig. 1g–i).

For the p-tau212 antibody, we performed additional specificity experiments using mass spectrometry on immunodepleted samples (IP-MS). The antibody did not bind to peptides phosphorylated at the epitopes threonine-181, serine-202, threonine-205, serine-214, threonine-217, and threonine-231. We observed binding to peptides phosphorylated at threonine-212 and double phosphorylated at threonine-212 and threonine-217. (Supplementary Table 1). Together, these results show that the p-tau212 and p-tau217 antibodies are specific to phosphorylation at the indicated positions.

### P-tau212 stains pathological tau structures in entorhinal cortex and hippocampal sections from AD human brains

To acknowledge if p-tau212 antibody recognizes neurofibrillary tangle pathology in AD brains, we performed immunohistochemistry in hippocampal and entorhinal cotex sections taken from $n = 1$ female (Case 2) and $n = 1$ male (Case 5) AD patients (Braak VI; duration of AD diagnosis = 6–9 years; age at death was 52–64 years). Strong staining of tau was observed in both cases (Fig. 2a, e). In the entorhinal cortex, neuropil threads, neurofibrillary tangles, and dystrophic neurites were observed (Fig. 2b, f). Higher magnification of the CA1 subfield of the hippocampus (Fig. 2c, g) and granule cell layer (GCL) (Fig. 2d, h) showed intense staining in the neurofibrylary tangles. Together, these results show that all major tau pathologies were stained by p-tau212 antibody.

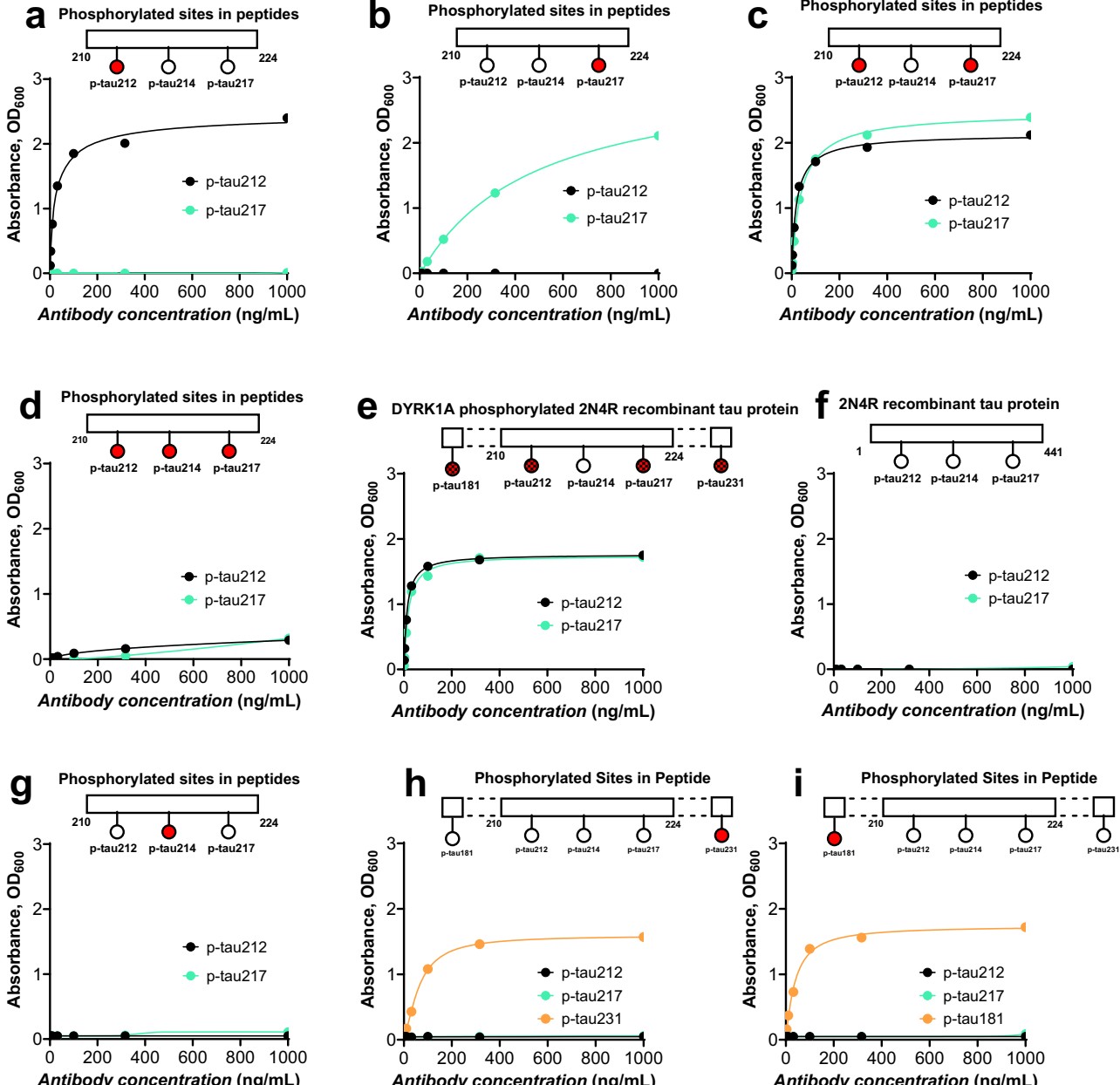

**Fig. 1 | Biochemical characterization of the p-tau212- and p-tau217-specific sheep mAbs.** Each panel shows the results of direct ELISAs where the p-tau212 and p-tau217 antibodies were each titrated against fixed concentrations of the indicated synthetic peptide or recombinant protein. The numbers at the start and the end of the horizontal schematics refer to the amino acids at the beginning and the end of the synthetic peptides. Colored circles represent phosphorylated site on the peptide. Checked circles represent epitopes proven to be phosphorylated by a dual specificity tyrosine phosphorylation regulated kinase 1A (DYRK1A) kinase.
**a** Binding profiles of the p-tau212 and p-tau217 antibodies to a synthetic peptide (Bt-x-SRpTPSLPTPPTREPK, where Bt-x refers to biotinylation) that was phosphorylated specifically and exclusively at threonine-212. **b** Kinetic profiles of the p-tau212 and p-tau217 antibodies to a synthetic peptide that had the same sequence as in (**a**) above but was rather phosphorylated only at threonine-217 (Bt-x-SRTPSLPpTPPTREPK). **c** Binding characteristics of the p-tau212 and p-tau217 antibodies to the same sequence of synthetic peptide as in (**a**) and (**b**) but was phosphorylated at both the threonine-212 and threonine-217 sites (Bt-x-

SRpTPSLPpTPPTREPK). **d** Binding profiles of the p-tau212 and p-tau217 antibodies to the same peptide sequence as in (**a**–**c**) except that it was phosphorylated jointly at threonine-212, serine-214 and threonine-217 (Bt-x-SRpTPpSLPpTPPTREPK). **e** Binding characteristics of the p-tau212 or p-tau217 antibodies to a recombinant form of full-length tau 441 (2N4R) that was phosphorylated in vitro by DYRK1A kinase. This kinase phosphorylates tau at multiple other sites beyond threonine-212 and threonine-217 but not at serine-214[29]. **f** Binding profiles of the p-tau212 or p-tau217 antibodies to a recombinant full-length tau 441 (2N4R) that was not phosphorylated at any site. **g** Binding characteristics of antibodies to peptide (bt-x-SRTPpSLPTPPTREPKK) specifically phosphorylated at serine-214. **h** Binding profile of antibodies to a peptide (bt-x-KKVAVVRpT(HOMOPRO)PKSPSSAK) specifically phosphorylated at threonine-231. P-tau231 specific antibody was used as a positive control. **i** Binding profile of antibodies to a peptide (bt-x-APKpTPPSSGE) specifically phosphorylated at threonine-181. P-tau181 specific antibody was used as a positive control. Source data are provided as a Source Data file.

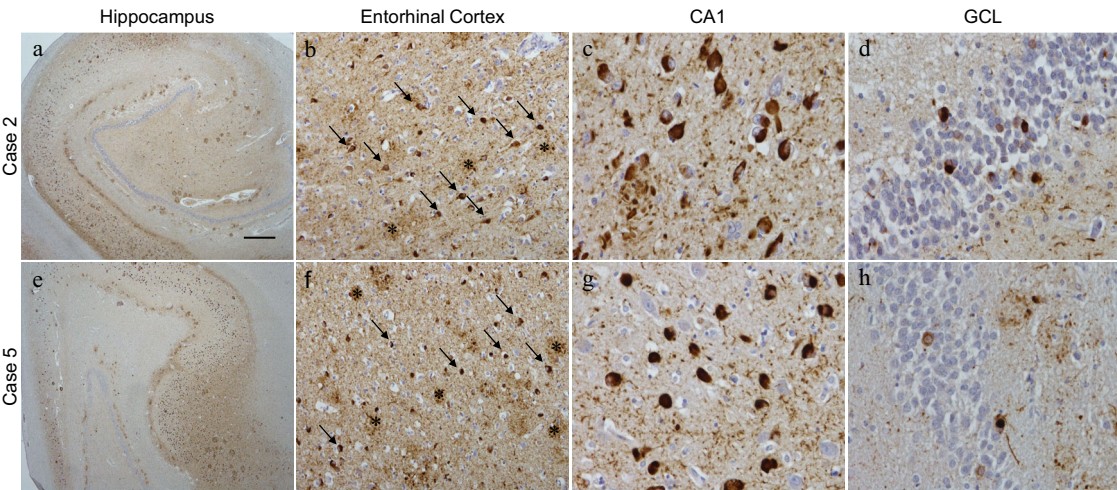

**Fig. 2 | P-tau212 immunohistochemical staining on Alzheimer's Disease cases.** Hippocampal sections were stained with the ptau212 antibody recognizing the phosphorylated residues at 212. In both cases 2 and 5, strong tau staining was observed (**a**, **e**). In the entorhinal cortex neuropil threads were seen covering the parenchyma, along with neurofibrillary tangles (**b** and **f** arrows) and dystrophic neurites observed in concentrated areas (**b** and **f** asterisks). Higher magnification images of the CA1 subfield of the hippocampus and the granule cell layer (GCL) show the intensity of the staining in the neurofibrillary tangles (CA1: **c** and **g**, GCL: **d** and **h**). Scale bar represents 500 μm in a and e; 50 μm in **b** and **f**; 25 μm in **c**, **d**, **g**, and **h**.

## P-tau212 and p-tau217 stain similar neurofibrillary tangle structures in AD human brain tissue

Next, we investigated if the p-tau212 antibody staining is comparable with the staining pattern of p-tau217. To achieve this, we performed immunofluorescent staining of tangles in brain slices from the temporal cortex of three different autopsy-verified AD patient brains (Braak VI; duration of AD diagnosis = 10–15 years; age at death was 58–68 years). For all three individuals, the p-tau212-specific antibody stained NFTs, including punctate pre-tangle-like aggregates as well as intracellular structures that colocalized with the nuclear stain DAPI (Fig. 3A). Importantly, the p-tau212 and p-tau217 antibodies both stained NFTs with high degrees of colocalization (Fig. 3A). Quantitative assessment revealed that 100% of the NFTs and 96% of neuropil threads were stained by both antibodies (Supplementary Table 2). Moreover, both antibodies covered similar areas of tangles and threads, slightly favouring p-tau217 for the former structures (Supplementary Fig. 2), and moderately favouring p-tau212 for the latter structures (Supplementary Fig. 3).

Furthermore, p-tau212 colocalized with AT8 (p-tau202/p-tau205) (Fig. 3B). The p-tau212 antibody stained 100% of the counted tangles, whereas AT8 stained 95% of these same NFTs. Additionally, p-tau212 and AT8 stainings were present in a similar number of neuropil threads (Supplementary Table 2). The stained area was similar for both antibodies, moderately in favor of p-tau212 in tangles (Supplementary Fig. 2) and slightly favorably for AT8 in threads (Supplementary Fig. 3). Together, these results indicate that p-tau212 and p-tau217 recognize similar tangle structures in autopsied AD brains, and that this is verified by colocalization with the commonly used NFT marker AT8.

## Development and validation of blood-based assay for p-tau212

We next developed an immunoassay to measure p-tau212 in plasma and CSF, by pairing the p-tau212 antibody with the N-terminal-tau targeting mouse monoclonal antibody Tau12 (BioLegend, #SIG-39416) against the epitope tau6-18. Following optimization of the biochemical parameters, we tested the assay's technical performance, following recommendations of an international consortium of clinical chemists[30]. The assay showed strong dilution linearity in both plasma and CSF (Supplementary Table 3); signals decreased proportionally when measured two- or four-fold diluted (for plasma; Supplementary Fig. 4a) and 16-, 32- or 64-fold (for CSF; Supplementary Fig. 4b).

Between-run % coefficients of variation (CV) for both matrices was 12.9–5.3% when two independent samples were measured in up to five separate analytical runs (Supplementary Fig. 4c, d). Follow-up precision experiment on de-identified plasma samples exhibited CVs <20% for 27/29 duplicates (Supplementary Fig. 5). Recovery of signal from exogenously added material was 85.3-103.1% for CSF (Supplementary Table 4) and 79.6–94.0% for plasma (Supplementary Table 5). The lower limit of quantification of (LLOQ) Simoa assay was 0.17 pg/ml. We additionally compared the performance of the plasma p-tau212 assay with a validated immunoprecipitation-mass spectrometry method for plasma p-tau217 measurement. We observed a strong correlation between the assays (R = 0.867; *P* < 0.0001) (Fig. 4). Development and validation of the plasma p-tau217 assay is described in a recent article[31].

## Participants

Studies included a total of *n* = 388 participants from five independent cohorts. Plasma samples from The Baltimore Longitudinal Study of Aging (BLSA)-Neuropathology cohort (*n* = 47; Supplementary Table 6) classified participants as autopsy-verified-AD (*n* = 20), asymptomatic-AD (ASYMAD) (*n* = 15) and non-AD controls (*n* = 12). CSF samples from The University of California San Diego (UCSD)-Neuropathology cohort (*n* = 67; Supplementary Table 7) included autopsy-confirmed high Alzheimer's disease neuropathologic change (ADNC) (*n* = 21), low ADNC (*n* = 8), other neuropathologies (*n* = 19) and mixed high-ADNC with other neuropathologies (*n* = 19). Plasma samples from The Gothenburg (*n* = 30; Supplementary Table 8) and plasma/CSF samples from the Polish (*n* = 95; Supplementary Table 9) cohorts consisted of individuals with abnormal CSF core biomarker profiles for AD and biomarker-negative controls. Plasma samples from the Slovenian memory clinic cohort from the University Medical Center, Ljubljana, (*n* = 149; Table 1) included patients in the AD continuum, that is, MCI due to AD (*n* = 41), and AD dementia (*n* = 62), as well as participants with subjective cognitive decline (*n* = 24) and MCI not due to AD (*n* = 22).

## Diagnostic performance of plasma p-tau212 to distinguish autopsy-verified AD from other neurodegenerative disorders

In the BLSA-Neuropathology cohort, plasma p-tau212 demonstrated high specificity for ADNC, being significantly higher in autopsy-verified AD compared with the ASYMAD and non-AD control groups (*P* < 0.0001; Fig. 5a for both). In comparison, plasma p-tau181 and

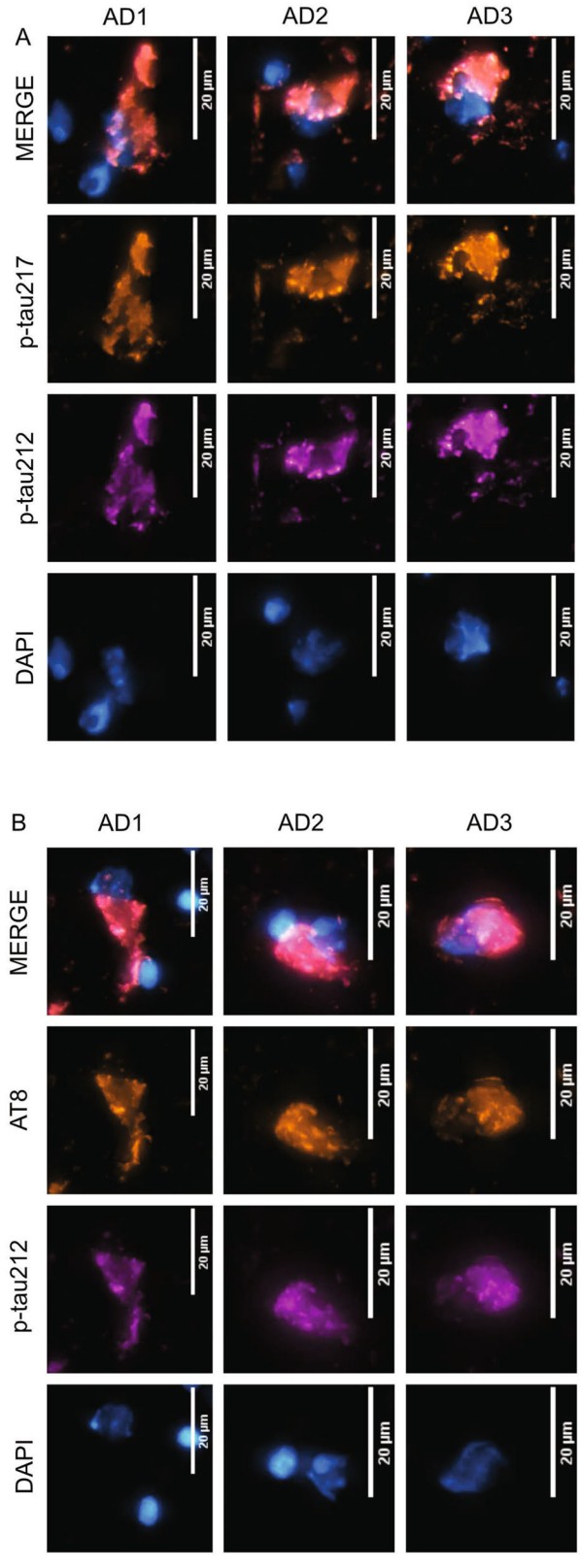

**Fig. 3 | Representative neurofibrillary tangles in autopsy-diagnosed AD brain tissue are phosphorylated jointly at p-tau212 and p-tau217.** The figure shows confocal microscopy images of immunofluorescent staining of selected tangles from temporal cortex tissue sections of three independent autopsy-verified AD patients. The individual was neuropathologically evaluated to be at Braak VI at autopsy, and they had been given an AD diagnosis for 15 years prior to death. Each micrograph shows co-staining with the nuclear stain DAPI in cyan hot, p-tau212 in magenta and p-tau217 (**A**, **b**) (or AT8 [p-tau202/205] in (**B**) in orange hot. The merged images show colocalization of the different color channels. The scale bar is 20 μm in each image.

and the Other Pathology groups ($p < 0.0063$ each; Fig. 5d). Moreover, those with mixed AD plus other pathologies (ADNC + Other) had significantly higher CSF p-tau212 versus both the Low Pathology and the Other Pathology groups ($p < 0.0179$ each; Fig. 5d), which was not the case for p-tau217, p-tau231 or p-tau181 (Supplementary Fig. 6). There was no difference between p-tau212 levels in those with ADNC with or without concomitant pathologies, and also between the Low and Other pathology groups ($p > 0.05$; Fig. 5d), indicating that the increases of p-tau212 were specific to ADNC.

In both cohorts, plasma/CSF p-tau212 had larger fold changes than the other p-tau biomarkers (Supplementary Table 10).

In the BLSA-Neuropathology cohort, plasma p-tau212 was higher in those with frequent versus sparse CERAD neuritic plaque scores[32] (Fig. 5b). For plasma p-tau181; however, we did not observe any statistically significant differences. Plasma p-tau231 was higher in the frequent versus sparse ($p = 0.03$) and the moderate versus frequent ($p = 0.02$) plaque score groups (Table 2).

In the UCSD-Neuropathology cohort, CSF p-tau212 was significantly increased in individuals with moderate plaque score ($p < 0.0083$, 6.1x estimated mean fold change) and frequent plaque score ($p < 0.0001$, 8.7x estimated mean fold change) versus those with sparse plaques (Fig. 5e). In comparison, CSF p-tau217, p-tau181 and p-tau231 were also significantly higher in individuals with moderate versus sparse plaque scores. However, the estimated mean fold increases were smaller (4.0x, 3.5x, 3.7x respectively) than for CSF p-tau212 (Table 2).

In the UCSD-Neuropathology cohort, CSF p-tau212 was significantly increased in individuals in Thal phase 5[33] compared with individuals in stages 0-2 ($p = 0.006$, 5.7x estimated mean fold change). Comparison of Thal phase 0-2 group with Thal phase 3-4 demonstrated 4.0x estimated mean fold increase in the latter, however, the result was not statistically significant ($p = 0.09$) (Fig. 5g). In comparison, CSF p-tau217, p-tau181 and p-tau231 were also increased in Thal phase 5 in reference to individuals in phases 0-2. CSF p-tau217 was also increased in comparison of Thal phases 0-2 and 3-4 (Table 2; Supplementary Fig. 6).

In the BLSA-Neuropathology cohort, plasma p-tau212 was significantly higher in Braak V-VI versus both Braak III-IV and I-II individuals ($p < 0.013$ each; Fig. 5c). Plasma p-tau231 performed similarly as p-tau212 but p-tau181 was not significantly different between any of the groups (Table 3).

In the UCSD-Neuropathology cohort, CSF p-tau212 tau was significantly different between Braak I-II and Braak V-VI; $p = 0.002$, 7.4x estimated mean fold change) and between Braak III-IV and Braak V-VI ($p = 0.0059$; 2.38x estimated mean fold change (Fig. 5f). CSF p-tau217 was increased in Braak V-VI versus Braak I-II ($p = 0.0002$, 4.6x estimated mean fold change) but not versus Braak III-IV ($p = 0.0685$; 1.6x estimated mean fold change; Table 3). CSF p-tau181 was also higher in Braak V-VI versus both Braak I-II ($p = 0.0021$, 4.4x estimated mean fold change) and Braak III-IV ($p = 0.0475$, 1.7x estimated mean fold change), (Table 3). Similar results were recorded for CSF p-tau231 (Table 3).

These results show that the levels of plasma p-tau212, similar to p-tau217, increase according to staging of ADNC.

p-tau231 were significantly higher in the AD versus non-AD control group ($p = 0.0033$ and $p = 0.0022$, respectively) but not between the AD and ASYMAD groups (Supplementary Table 10). Plasma p-tau217 data was unavailable for this cohort.

In the UCSD-Neuropathology cohort, CSF p-tau212 was significantly higher in the ADNC group versus each of the Low Pathology

**Correlation of Simoa p-tau212 with mass spectrometry p-tau217**

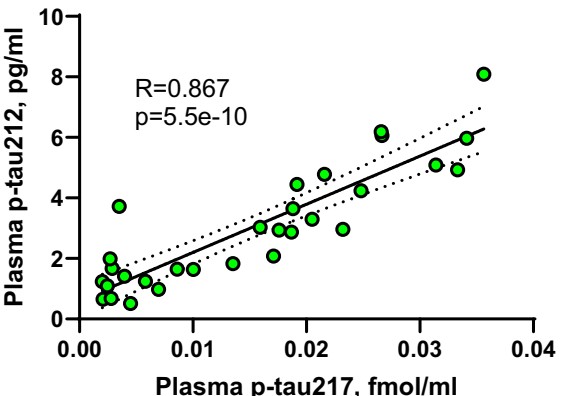

R=0.867
p=5.5e-10

**Fig. 4 | Spearman Correlation for IP-MS plasma p-tau217 and Simoa plasma p-tau212.** The figure shows correlation between IP-MS plasma p-tau217 and Simoa plasma p-tau212 ($n = 30$ for both). Spearman $r = 0.867$. $p = 5.5$-10.

### Plasma p-tau212 comparison with plasma p-tau181 and p-tau231 to differentiate Aβ + AD from Aβ- controls

In the Gothenburg cohort, plasma p-tau212 was 3.7-fold higher in biomarker-positive AD versus biomarker-negative controls ($p < 0.0001$; Fig. 6d), which was higher than fold increases of 1.9 for p-tau181 and 2.3 for p-tau231 ($p < 0.0003$ each). Plasma p-tau212 had a diagnostic accuracy of 91.5% (95% CI = 79.1%–100%), being numerically higher than 86.6% (95% CI = 72.3%–100%) for p-tau181 and 86.6% (95% CI = 70.7%–100%) for p-tau231 (Fig. 6e). DeLong test comparison of plasma p-tau212 with p-tau181 and p-tau231 comparison showed no significant differences ($p = 0.6107$ and $p = 0.5492$ respectively).

### Comparison of plasma versus CSF p-tau212 to differentiate Aβ + AD from Aβ- controls

In the Polish cohort with paired plasma and CSF samples, plasma p-tau212 was 4.0-fold higher in biomarker-positive AD versus biomarker-negative controls ($p < 0.0001$; Fig. 6a), whilst CSF p-tau212 was 10.0-fold higher ($p < 0.0001$; Fig. 6b). Plasma p-tau212 had a diagnostic accuracy of 85.6% (95% CI = 75%–95.7%) which was statistically not different from that of CSF p-tau212 (91.9%, 95% CI = 83.4%–100%; Fig. 6), DeLong test ($p = 0.2233$).

Together, plasma and CSF p-tau212 have comparable performances to separate biomarker-positive AD versus biomarker-negative controls.

### Plasma p-tau212 versus p-tau217 in a memory clinic cohort

In the Slovenia cohort, both plasma p-tau212 and p-tau217 levels demonstrated stepwise increases from SCD and non-AD groups to AD-dementia (Fig. 7a, b). We used the clinical diagnostic groups assigned to these patients, in addition to their Aβ positivity, to ascertain the biomarker changes across disease stages. Of note, a CU group was lacking in this clinical population including only participants who reported cognitive symptoms. For both markers, the lowest concentrations were in the Aβ- SCD and Aβ- non-AD MCI (Fig. 7a, b).

Plasma p-tau212 and p-tau217 had statistically not different AUCs to distinguish Aβ- SCD from Aβ + AD-dementia (AUC = 92.5% [95% CI = 87.0%–97.9%] versus 95.6% [95% CI = 91.7%–99.4%]; DeLong test $p = 0.232$, Fig. 7d) and to separate Aβ + AD-dementia from Aβ- non-AD MCI (AUC = 89.2% [95% CI = 88.2%–96.2%] versus AUC = 87.8% [95% CI = 78.2%–97.2]; De Long test $p = 0.8119$; Fig. 7e). Positive predictive value (PPV) for Aβ positivity versus SCD were 94.7% for p-tau212 and 98.1% for p-tau217. The negative predictive values (NPVs) were 72.4% and 74.2% respectively.

Next, we compared the concordance of plasma p-tau212 and p-tau217 in correctly identifying Aβ-positivity among the memory clinic participants. Aβ positivity was defined as CSF Aβ42/40 ratio <0.077 pg/ml, a cut-off value used in the Slovenian hospital. We dichotomized p-tau212 and p-tau217 values by generating within-cohort cutoffs of 3.2 pg/ml and 4.4 pg/ml, using the Youden's index approach. Plasma p-tau212 and p-tau217 concentrations had a high degree of agreement (83.5%), which included 34.1% (mostly Aβ- SCD individuals) and 49.5% (principally Aβ + AD dementia individuals) with normal and abnormal profiles, respectively, of both biomarkers.

These findings indicate that plasma p-tau212 and p-tau217 share a high degree of agreement to identify abnormal CSF Aβ42/40 results.

### Plasma p-tau212 and p-tau217 associations with Aβ and tau pathophysiology

In the Slovenian cohort, plasma p-tau212 and p-tau217 had comparable correlations with each of CSF Aβ42/Aβ40 ratio (Spearman's rho = −0.48 versus −0.54, $p < 0.0001$), CSF p-tau181 (Spearman's rho = 0.51 versus 0.55, $p < 0.0001$), and CSF total-tau (Spearman's rho = 0.55 versus 0.58, $p < 0.0001$) measured with the Innotest assays. Similarly equivalent correlations of p-tau212 and p-tau217 with Aβ and tau at neuropathology were observed (Supplementary Table 11).

Plasma p-tau212 and p-tau217 had comparable correlations with in vivo tau pathology at autopsy assessed with Braak staging in the UCSD-Neuropathology cohort (Spearman's rho = 0.67, versus = 0.59, $p < 0.0001$) (Supplementary Table 11). Furthermore, plasma/CSF p-tau212 and p-tau217 showed comparable correlations with other biofluid biomarkers in the different cohorts (Supplementary Table 11).

### Association of p-tau212 and p-tau217 with cognition

P-tau212 in CSF was inversely correlated with MMSE in the Polish and UCSD-Neuropathology cohorts (Spearman rho = −0.42, $p < 0.0001$; rho = −0.3, $p = 0.012$), respectively. Plasma p-tau212 was correlated inversely with MMSE in the Polish cohort (Spearman rho = −0.49; $p < 0.0001$). Similar results were obtained in the Slovenian cohort for both p-tau212 and p-tau217 (Spearman rho = −0.338, $p = 0.0005$ versus −0.222, $p = 0.025$ respectively).

## Discussion

In the present study, we show that p-tau212 serves as a biochemical marker for AD. The p-tau212 mAb specifically reacted with p-tau212-containing peptides, but not with peptides phosphorylated exclusively at positions threonine-217, serine-214 or other neighboring phosphorylation sites. Further, the p-tau212 antibody showed strong immunostaining of NFTs. A well-validated plasma p-tau212 Simoa method developed in this study was increased in antemortem blood samples from individuals with autopsy-verified ADNC, correlated with amyloid plaque and NFT densities, and could differentiate AD from other neurodegenerative disorders. Furthermore, plasma p-tau212 showed indifferent diagnostic performances as CSF p-tau212 to separate Aβ + AD dementia participants from Aβ- controls. In a memory clinic population, plasma p-tau212 levels were highest in AD dementia and lowest in those with SCD, indicating that phosphorylation at this epitope increases with disease severity, similar to what has been found for other p-tau epitopes[8–10]. Further, p-tau212 had generally higher diagnostic accuracies and larger fold changes than p-tau181 and p-tau231. Importantly, the results for plasma p-tau212 had high degrees of concordance with those for plasma p-tau217, suggesting similarities in disease-associated phosphorylation at these two separate sites that are located in the same short domain in the proline-rich region of tau.

Previous reports have shown that phosphorylation at threonine-212 inhibits tau protein's physiological function of binding to microtubules, promotes tau self-assembly to form aggregates, and

**Table 1 | Demographic characteristics of the Slovenia memory clinic cohort**

|  | SCD | Non-AD MCI | AD-MCI | AD-Dementia |
|---|---|---|---|---|
| Sample size | 24 | 22 | 41 | 62 |
| Age, y | 64.30 ± 8.51[c,d] | 67.67 ± 13.07[c,d] | 75.46 ± 5.71[a,b] | 75.22 ± 7.46[a,b] |
| Sex, F, *n (%)* | 13/24 (54.2%) | 12/22 (54.6%) | 26/41 (63.4%) | 39/62 (62.9%) |
| MMSE | 28 (26.25–29) | 25 (24.8–26) | 27 (25–29) | 21 (17–25)[a,c] |
| CSF Aβ42, pg/ml | 1270 (1120–1465)[c,d] | 1341 (1082–1489)[c,d] | 696 (606.5–949.5)[a,b] | 623.5 (550–751.8)[a,b] |
| CSF Aβ42/Aβ40 ratio | 1.30 (0.11–0.14)[c,d] | 0.115 (0.11–0.13)[c,d] | 0.061 (0.49–0.07)[a,b] | 0.048 (0.038–0.058)[a,b] |
| CSF total-tau, pg/ml | 200 (181.0–291)[c,d] | 231 (198.3–304.8)[c,d] | 391 (333–584.5)[a,b,d] | 815 (645–1090)[a,b,c] |
| CSF p-tau181 (Innotest), pg/ml | 38 (33.5–48)[c,d] | 43.5 (37.5–50.3)[c,d] | 64 (55–101.5)[a,b,d] | 111 (92.3–140.5)[a,b,c] |
| Plasma p-tau212, pg/ml | 1.750 (0.9–2.8)[c,d] | 1.850 (1.3–3.6)[c,d] | 3.4 (2–5)[a,b,d] | 5.9 (4–7.8)[a,b,c] |
| Plasma p-tau217, pg/ml | 1.7 (0.8–3)[c,d] | 2.1 (1.4–3.8)[c,d] | 5.1 (3.3–7.8)[a,b,d] | 8.1 (5.9–10.6)[a,b,c] |

[a]Statistically different from SCD.
[b]Statistically different from non-AD MCI.
[c]Statistically different from AD-MCI.
[d]Statistically different from AD-Dementia.

demonstrates cytotoxic effects[24,29]. Threonine-212 was found to be a target for kinases that do not phosphorylate threonine-217[27,28]. A proteomic study found that p-tau212 is prominent in AD brain tissue compared with controls, with a similar degree of phosphorylation as p-tau217[34]. These findings implicate p-tau212 as an important candidate biomarker of AD pathophysiology and neuropathology. We generated a highly specific sheep mAb for this site. ELISA assays demonstrated specificity and lack of cross-reactivity to p-tau217 peptides, which contrasts other plasma p-tau217 assays which use antibodies that are non-selective for the threonine-212 and threonine-217 sites[21,35]. For this reason, the p-tau212-specific antibody allowed us to delineate the biomarker functions of p-tau212, independent of phosphorylation at neighboring sites. The presence of the p-tau212 on NFTs from autopsy-verified AD brain tissues, which recognized similar structures as did p-tau217 and p-tau202/205, suggests that NFTs are phosphorylated at threonine-212. In the biological course of AD, increasing fractions of p-tau212-containing tau forms might be released into biofluids, including CSF and blood, which explains the association of plasma/CSF p-tau212 concentrations retrospectively measured in antemortem samples with NFT content at autopsy. Our results agree with the hypothesis that the tau forms that become available in blood early in AD consist of N-terminal and mid-region fragments that would include the threonine-212 site[2]. Additionally, the association of plasma p-tau212 with Aβ plaques – at autopsy as well as with antemortem CSF Aβ42/Aβ40 ratio – especially in individuals with MCI, indicates that the release of tau variants containing p-tau212 is an early pathological event. Interestingly, we did not observe an increase of p-tau levels in the ASYMAD group. Amyloid, and tau related changes in the hippocampus at autopsy in this sub-cohort were not different from MCI, and higher than in the control group, but ASYMAD patients do not demonstrate cognitive decline[36]. The finding of normal p-tau212 levels in the ASYMAD group suggests a connection of p-tau release into the bloodstream with functional impairment of neurons. Normal levels of p-tau might suggest resilience to dementia stages, despite neuropathological AD manifestations. The specific increases of plasma p-tau212 in autopsy-verified AD patients and in AD but not in other neurodegenerative diseases, indicate that the molecular events that lead to the availability of tau hyperphosphorylated at this site in the blood is exclusive to AD. These increases were also evident in individuals with mixed pathology.

The similar performance of p-tau212 between plasma and CSF to separate AD dementia participants from Aβ-negative controls suggest that measurement of this biomarker may have a place in the clinical evaluation of patients with suspected AD. This would lead to significant savings in analysis time and cost while improving the acceptability of biomarker measures for communities who either do not have the resources or would not consent to lumbar puncture. Besides p-tau212, plasma p-tau217 is the other p-tau biomarker that has been shown to have high associations with AD similar to CSF; for p-tau181 and p-tau231, diagnostic accuracies were higher in CSF than in plasma[9,20]. This finding should be replicated in other independent cohorts.

To ascertain if phosphorylation within the same stretch of proline-rich amino acids share similarities in their pathological roles in AD brains and the release of p-tau-containing tau forms into blood, we compared the cellular localization of p-tau212 in brain NFTs and its diagnostic performances in plasma with those of plasma p-tau217. We found co-staining of p-tau212 and p-tau217 in 100% of tangles and 96% of neuropil threads, with high degree of the areas covered in all three brains.

The similar patterns of staining and colocalization on NFTs in brain tissue, in addition to the high degree of agreement of plasma p-tau212 and p-tau217 to identify a positive CSF Aβ42/40 ratio, strongly suggest that these two sites can be used as separate biomarkers, but used together could provide complementary information about disease stage, when measured in the same patient. Consequently, similar fractions of CNS-derived species of p-tau212 and p-tau217 may be released into blood at defined stages of AD, which explains their resemblant diagnostic performances. Furthermore, plasma p-tau212 and p-tau217 both had similarly large fold changes between diagnostic groups that exceeded those for plasma p-tau181 and p-tau231. Large fold difference between diagnostic groups is a desirable characteristic of a robust biomarker that is important for minimizing day-to-day variability[7]. The plasma p-tau212 assay presented in this study stand to fill a critical gap in access to high-performing blood tests.

The strengths of this study include the generation and biochemical characterization of p-tau212- and p-tau217-specific mAbs, and analyzes of their immunohistochemical staining patterns on AD brain tissue slices. Others include the development and technical validation of the plasma p-tau212 assay. Additionally, we found high diagnostic accuracies of these assays in five independent cohorts, including blood-to-autopsy and memory clinic cohorts. We also compared the diagnostic performances of plasma p-tau212 with those of CSF p-tau212 as well as plasma p-tau181, p-tau231, and p-tau217 in various cohorts, where plasma p-tau212 has excellent performance.

Limitations of this study include lack of separate Braak and Thal phase 0 negative control group. Additionally, the Polish cohort, which was used for the comparison of p-tau212 immunoassay performance in plasma and CSF consisted of *n* = 95 participants, and verification of the assay performance in larger cohorts would be favorable.

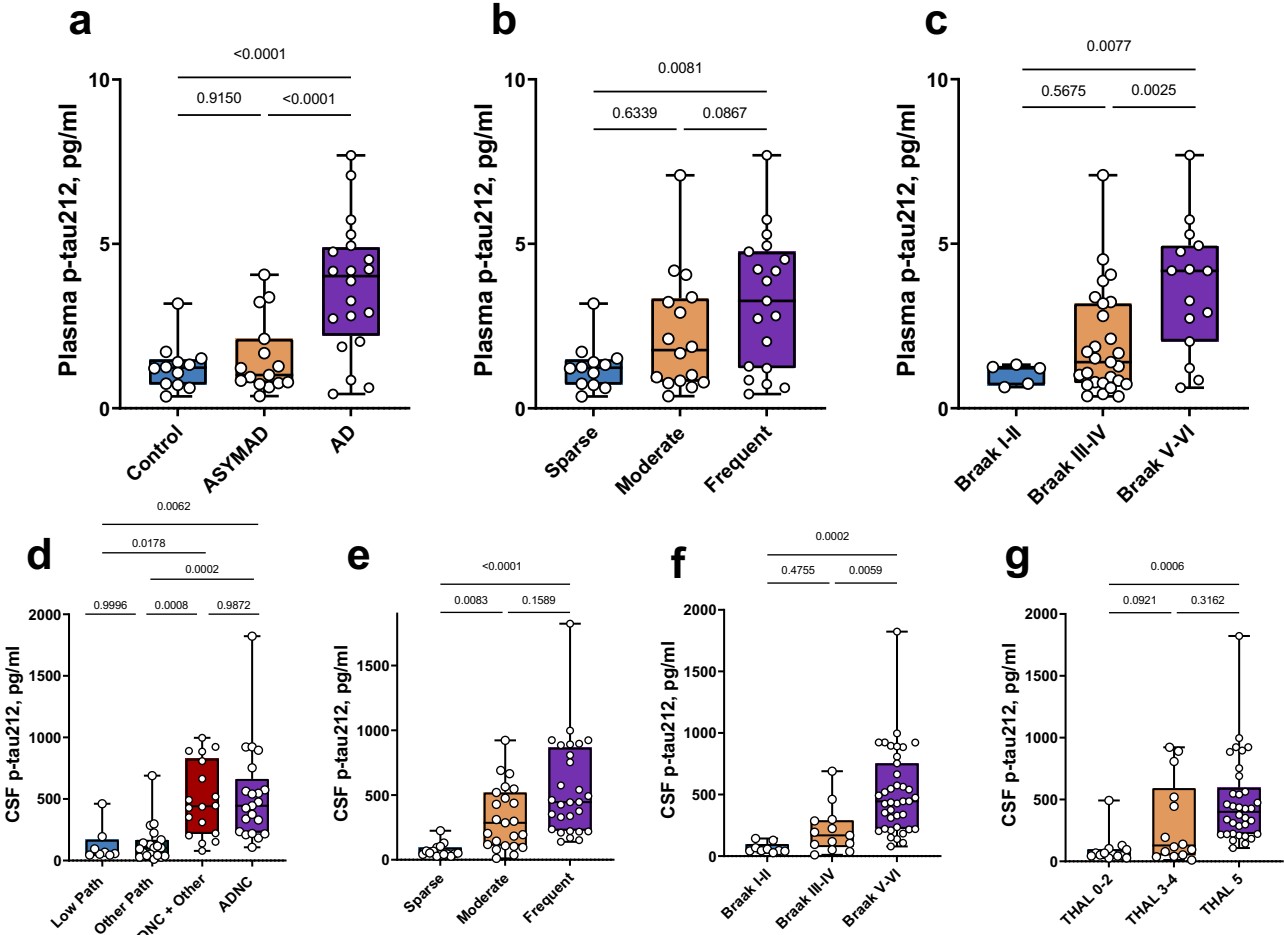

**Fig. 5 | Clinical performance of plasma and CSF p-tau212 in autopsy verified cohorts.** The figure shows plasma and CSF p-tau212 levels according to diagnostic groups, amyloid pathology, and Braak staging of neurofibrillary tangles in the BLSA- (a-c) and UCSD-neuropathology (d-f) cohorts with post-mortem validation. **a** Plasma p-tau212 levels in the control ($n = 12$) ASYMAD ($n = 15$), and AD ($n = 20$) groups in the BLSA-neuropathology cohort. **b** Plasma p-tau212 concentrations according to Aβ plaque counts categorized by the CERAD scoring – Sparse ($n = 12$), Moderate ($n = 16$) and Frequent ($n = 19$). **c** Stepwise increase of plasma p-tau212 levels according to Braak staging of neurofibrillary tangles - Braak I-II ($n = 5$), Braak III-IV ($n = 27$), Braak V-VI ($n = 15$). **d** CSF p-tau212 levels according to Alzheimer's disease neuropathologic change (ADNC) categorization. The groups include those with low ADNC pathology - Low Path (non ADNC) ($n = 8$), Other Path ($n = 19$), ADNC ($n = 21$) and ADNC with concomitant neurodegenerative pathologies (ADNC + Other) ($n = 18$). **e** CSF p-tau212 concentrations according to the CERAD scores of Aβ plaques Sparse ($n = 15$), Moderate ($n = 23$), Frequent ($n = 28$) **f** CSF p-tau212 levels separated based on Braak staging characterization given at autopsy - Braak I-II ($n = 9$), Braak III-IV ($n = 13$), Braak V-VI ($n = 39$). **g** CSF p-tau212 concentration across different Thal phases - THAL 0-2 ($n = 13$), THAL 3-4 ($n = 14$); THAL 5 ($n = 34$). The estimated mean between-group fold differences of CSF/plasma p-tau212 for every plot are given in Tables 2, 3, and Supplementary Table 10. Comparisons of the performances of p-tau212 with p-tau217, p-tau181 and p-tau231 are shown in Tables 2, 3, and Supplementary Table 10. Boxplots showing measurements of other p-tau biomarkers are shown in Supplementary Fig. 6. Box plots are shown as median and interquartile range (IQR), boundaries of the whiskers are minimum and maximum values. Analysis of variance (ANOVA) with Tukey's post-hoc test was used to compare differences between groups, after adjusting for sex, age, and CSF/plasma collection-to-death intervals. Pairwise comparisons were adjusted for multiple comparisons. Source data are provided as a Source Data file.

In conclusion, p-tau212 is a promising biomarker, tightly associated with AD pathophysiology and neuropathology. NFTs in AD brains were phosphorylated at this epitope, and an ultra-sensitive biomarker developed to quantify p-tau212 in blood showed similar performances to CSF p-tau212, and also plasma p-tau217. Plasma p-tau212 was associated with Aβ and tau pathology evaluated at autopsy and in memory clinic settings using CSF assays. The specificity of this biomarker to AD pathogenesis was further supported by the similarly increased levels in older adults with AD with or without mixed neurodegenerative pathologies. These results show that plasma p-tau212 tracks AD pathophysiology across the disease continuum, and thus has the potential for clinical diagnostic, prognostic, and population screening purposes, as well as to identify and longitudinally monitor memory clinic patients who are eligible for anti-AD therapies. Additionally, the close proximity of p-tau212 to p-tau217 makes it very promising target that could be longitudinally associated with disease progression and conversion from

cognitively unimpaired (CU) people to dementia. Moreover, p-tau212 needs to be validated against low-threshold centiloid positivity in CU cohort, to verify its utility in the pre-symptomatic phases. Furthermore, p-tau212 levels were proven to be increased in AD-DS brains[37], therefore this biomarker might find utility as a biomarker for this population.

## Methods

The research complies with all relevant ethical regulations: National Institute of Health (IRB 03AG0325); University of Gothenburg (#EPN140811); University of Wroclaw (KB-380/2017); University of California San Diego (IRB 170957), Medical Ethics Committee of the Republic of Slovenia, Ministry of Health (0120-342/2021/6).

### Antibody development

Generation of sheep monoclonal antibodies for p-tau212 and p-tau217 was undertaken according to the UK Animal Scientific Procedures Act,

**Table 2 | Comparison of the estimated mean fold changes of different p-tau variants between different stages of Amyloid pathology in the post-mortem cohorts**

| BLSA-Neuropathology cohort: amyloid pathology | | | | | |
|---|---|---|---|---|---|
| Test - Plasma | Sparse versus Moderate | | Sparse versus Frequent | | Moderate versus Frequent |
| Result/analyte | Fold change | P-value | Fold change | P-value | Fold change | P-value |
| p-tau181 | 1.39 | 0.6703 | 1.69 | 0.2107 | 1.22 | 0.7231 |
| p-tau231 | 1.00 | 0.9999 | 1.70 | 0.0304 | 1.70 | 0.0271 |
| p-tau212 | 1.55 | 0.6339 | 2.70 | 0.0081 | 1.74 | 0.0867 |
| UCSD-Neuropathology cohort: CERAD amyloid pathology | | | | | |
| –Test - CSF | Sparse versus Moderate | | Sparse versus Frequent | | Moderate versus Frequent |
| Result/analyte | Fold change | P-value | Fold change | P-value | Fold change | P-value |
| p-tau181 | 3.46 | 0.0114 | 4.21 | 0.0002 | 1.21 | 0.5582 |
| p-tau231 | 3.73 | 0.0122 | 4.57 | 0.0003 | 1.23 | 0.5536 |
| p-tau212 | 6.06 | 0.0083 | 8.72 | <0.0001 | 1.43 | 0.1589 |
| p-tau217 | 3.96 | 0.0034 | 5.27 | <0.0001 | 1.33 | 0.2053 |
| UCSD-Neuropathology cohort: Thal amyloid pathology | | | | | |
| –Test - CSF | Thal 0-2 versus Thal 3-4 | | Thal 0-2 versus Thal 5 | | Thal 3-4 versus Thal 5 |
| Result/analyte | Fold change | P-value | Fold change | P-value | Fold change | P-value |
| p-tau181 | 2.87 | 0.0799 | 3.47 | 0.0031 | 1.21 | 0.6684 |
| p-tau231 | 3.04 | 0.0781 | 3.62 | 0.0039 | 1.19 | 0.7355 |
| p-tau212 | 4.02 | 0.0921 | 5,73 | 0.0006 | 1.43 | 0.3162 |
| p-tau217 | 3.38 | 0.0403 | 4.55 | 0.0002 | 1.35 | 0.2930 |

**Table 3 | Comparison of the estimated mean fold changes of different p-tau variants between different stages of Braak tangle pathology in the post-mortem cohorts**

| BLSA-Neuropathology cohort: tangle pathology | | | | | |
|---|---|---|---|---|---|
| –Test - Plasma | Braak I-II versus Braak III-IV | | Braak I-II versus Braak V-VI | | Braak III-IV versus Braak V-VI |
| Result/analyte | Fold change | P-value | Fold change | P-value | Fold change | P-value |
| p-tau181 | 1.04 | 0.9951 | 1.66 | 0.3631 | 1.59 | 0.1052 |
| p-tau231 | 1.15 | 0.9036 | 2.16 | 0.0129 | 1.88 | 0.0004 |
| p-tau212 | 1.91 | 0.5675 | 3.98 | 0.0077 | 2.08 | 0.0025 |
| UCSD-Neuropathology cohort: tangle pathology | | | | | |
| –Test - CSF | Braak I-II versus Braak III-IV | | Braak I-II versus Braak V-VI | | Braak III-IV versus Braak V-VI |
| Result/analyte | Fold change | P-value | Fold change | P-value | Fold change | P-value |
| p-tau181 | 2.43 | 0.4392 | 4.45 | 0.0021 | 1.69 | 0.0475 |
| p-tau231 | 2.21 | 0.5179 | 4.24 | 0.0018 | 1.92 | 0.0383 |
| p-tau212 | 3.12 | 0.4755 | 7.44 | 0.0002 | 2.38 | 0.0059 |
| p-tau217 | 2.86 | 0.1819 | 4.63 | 0.0002 | 1.62 | 0.0685 |

and the methodology has been described previously[38]. Briefly, custom-designed p-tau peptides phosphorylated at threonine-212 and threonine-217 with C-terminal tetanus toxin sequences were synthesized (Severn Biotech, UK). These peptides were used for the immunization of sheep and the monoclonal antibody generation process followed as described[38]. Afterwards, candidate hybridomas were selected based on binding to specifically phosphorylated peptides, as described in Fig. 1. The p-tau212 and p-tau217 clones demonstrated the highest level of specificity for the p-tau212 and p-tau217 epitopes, respectively. Antibody design, generation and validation were performed at Bioventix Plc (Surrey, United Kingdom).

## Characterization of the p-tau212 and p-tau217 antibodies using enzyme-linked immunosorbent assays (ELISAs)

To validate antibody specificity, each antibody was incubated with a series of different p-tau peptides or intact 2N4R recombinant tau protein. For p-tau peptide analysis, a 96-well plate was coated with streptavidin (1000 ng/ml) and subsequently used to immobilize p-tau biotinylated peptide (500 ng/ml). For intact 2N4R recombinant tau proteins, each plate was directly coated (500 ng/ml). The p-tau212 and p-tau217 antibodies were applied at a range of concentrations (0 to 1000 ng/ml). Binding of the p-tau antibodies to immobilized peptide/protein was determined using a secondary anti sheep IgG HRP linked antibody. Following incubation with the secondary antibody the reaction was stopped by the addition of 1 M phosphoric acid and the subsequent color change was monitored by reading the absorbance of 450 nm ($OD_{450}$).

## Immunohistochemistry and fluorescence microscopy

For the single p-tau212 immunohistochemistry a total of 8 μm paraffin-embedded formalin-preserved tissue sections were cut from the hippocampus. Routine IHC was performed on sections using p-tau212 antibody. Briefly tissue sections were de-waxed in xylene and rehydrated through various alcohol concentrations, then pre-blocked using methanol and hydrogen peroxide to prohibit endogenous peroxidase activity. Antigen retrieval was carried out by pressure cooking slides in pH 6.0 citrate buffer. A solution of 10% milk/TBS-T was used to prevent non-specific antigen/antibody binding. Tissue sections underwent incubation p-tau212 antibody (1:100) for 1 h at room temperature, followed by biotinylated anti-sheep secondary antibody (1:100) and finally avidin-biotin complex (ABC), both for 30-min incubations. 3,3′-di-aminobenzidine (DAB) was used to develop color. Counterstaining was accomplished using Mayer's haematoxylin.

For immunohistochemistry and fluorescent tau imaging, human brain sections on superfrost glasses were fixed in a gradient concentration of ice cold 95%, 70% ethanol and 1X phosphate buffered saline (PBS) at room temperature. Sections were then blocked with 0.1% Bovine Serum Albumin (BSA; Sigma Aldrich #SLCH3826) in 0.1% Triton in Phosphate buffered saline (PBST) for 90 min at room temperature. The sections were then incubated with a cocktail of the following primary antibodies diluted in antibody diluent (0.1% BSA in 0.2% PBST) for 18 h at 4 degree celsius: AT8 (Thermo Fisher Scientific #MN1020), p-tau212 (Bioventix), and p-tau217 (Thermo Fisher; catalog # 44-744). AT8 was diluted at 1:500, p-tau212 at 1:2000 and p-tau217 at 1:1000. The use of antibodies generated in different host systems (AT8 in mouse, p-tau212 in sheep and p-tau217 in rabbit) was necessary for the selection of unique secondary antibodies in this experiment. The AT8 and p-tau217 antibodies have been biochemically validated previously[39,40]. Sections were washed with 0.1% PBST and incubated with the following secondary antibodies from Thermo Fisher Scientific (anti-sheep Alexa-fluor 488, #2432029; anti-rabbit Alexa-fluor 647, #1741783; and anti-mouse Alexa-fluor 647, #2170302) for 60 min at room temperature. All the brain sections were then treated with autofluorescence quenching agent TrueBlack™ 1X for 30 s and subjected to three 1X PBS washes of 5 min each. Lastly, all the sections were incubated with DAPI (Thermo Scientific #D1306) for nuclear staining followed by a washing step with 1X PBS. All the sections were mounted with DAKO fluorescent mounting media and incubated in the dark for 24 h before imaging.

The multichannel imaging of immuno-stained human brain sections was performed using an automatic widefield microscope (Axio Observer Z1, Zeiss, Germany). Multi-channel images were captured using DAPI, Alexa-fluor 488 and Alexa-fluor 647 filter sets.

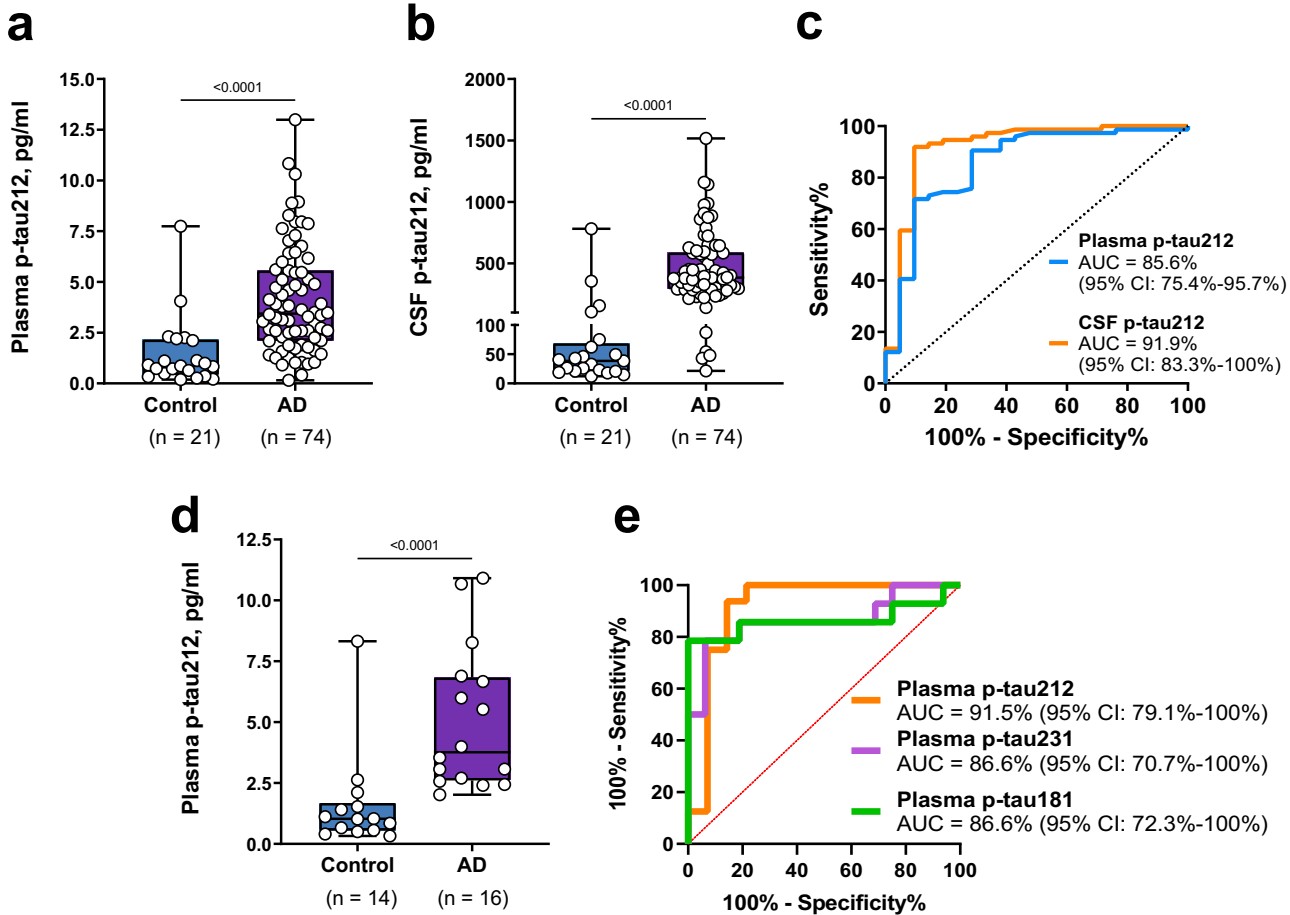

**Fig. 6 | P-tau212 performance in plasma versus CSF to distinguish Aβ + AD from Aβ- controls.** The plots in **a** and **b** show p-tau212 levels in Aβ-positive AD (n = 74) participants compared with Aβ-negative controls (n = 21) in paired plasma and CSF samples respectively in the Polish cohort. **c** AUCs of the accuracies of p-tau212 in plasma versus CSF to separate the groups shown in (**a**) and (**b**). De Long's test comparing the AUCs for plasma and CSF p-tau212 did not reach statistical significance. **d** Plasma p-tau212 levels in Aβ-positive AD (n = 16) participants compared with Aβ-negative controls (n = 14) in the Gothenburg cohort. **e** Comparison of the AUC values for plasma p-tau212 versus those for plasma p-tau231 and p-tau181 in

the same set of samples shown in (**d**). Box plots are shown as median and inter-quartile range (IQR), boundaries of the whiskers are minimum and maximum values. Group differences were examined using two-tailed Mann–Whitney test. The Polish cohort included individuals with paired CSF and EDTA plasma samples (n = 95) from Erlangen Score defined controls (n = 21) and AD (n = 74). The Gothenburg discovery cohort (n = 30) included EDTA plasma samples from neurochemically defined Alzheimer's disease participants (n = 16) and controls (n = 14). Source data are provided as a Source Data file.

All the images were captured using Plan-Apochromat 20X/0.8 DIC air objective lens. The acquisition settings were adjusted to prevent saturation or bleed through during the image acquisition. The FIJI ImageJ software was used to assign pseudo/LUT color for representation; cyan hot for DAPI, magenta for Alexa-fluor 488 (p-tau212), and orange hot for Alexa-fluor647 (p-tau217 or AT8). Multichannel image files from each channel were split into gray scale images and were subjected to background subtraction (rolling value = 50 pixels). A size exclusion criterion using 'Analyze Particle' was applied on the segmented images for tangle count (40 µm²–300 µm², circularity =;0.00-Infinity) and for neuropil threads (0 µm²–40 µm²) with circularity index of (circularity = 0.00–0.50; to pick up elongated threads). Any nonspecific segmentation for example from Aβ plaques/dystrophic neurites if any were excluded from the analysis. A summary of the results was used for quantification of the data.

**Plasma p-tau212 immunoassay development and clinical studies**
For the p-tau212 assay, which was developed on the Simoa HD-X instrument (Quanterix, MA, USA), the p-tau212 antibody was used as the capture antibody. A mouse monoclonal antibody raised against the N-terminal region of tau (Tau12; BioLegend, #SIG-39416) was used for

detection. In vitro phosphorylated recombinant full-length tau-441 (#269022, Abcam) was used as the assay calibrator. Blood samples and calibrators were diluted with assay diluent (Tau 2.0 Sample Diluent; #101556, Quanterix).

Analytical validation of the p-tau212 assay followed protocols described previously[8,10,31,39]. Assay development work was conducted at the University of Gothenburg, Sweden. In the clinical studies, p-tau212 was measured at the University of Gothenburg by scientists blinded to participant information, using the above-described assays. For p-tau212, plasma and CSF samples were diluted 1.2 times and 10 times respectively prior to measurement.

**Plasma mass spectrometry**
EDTA plasma samples (1 ml) were thawed, vortexed for 30 s at 2000 r.p.m. and spun down for 10 min at 4000 g. Tau protein was extracted by IP using beads (Dynabeads M-280 sheep anti-mouse IgG, Thermo Fisher Scientific, 11202D) cross-linked with a combination of antibodies targeting non-phosphorylated tau: Tau12 (BioLegend, 806501), HT7 (Thermo Fisher Scientific, MN1000) and BT2 (Thermo Fisher Scientific, MN1010). Antibodies were conjugated to the beads at a concentration of 4 µg antibody/50 µl beads. Automated IP was

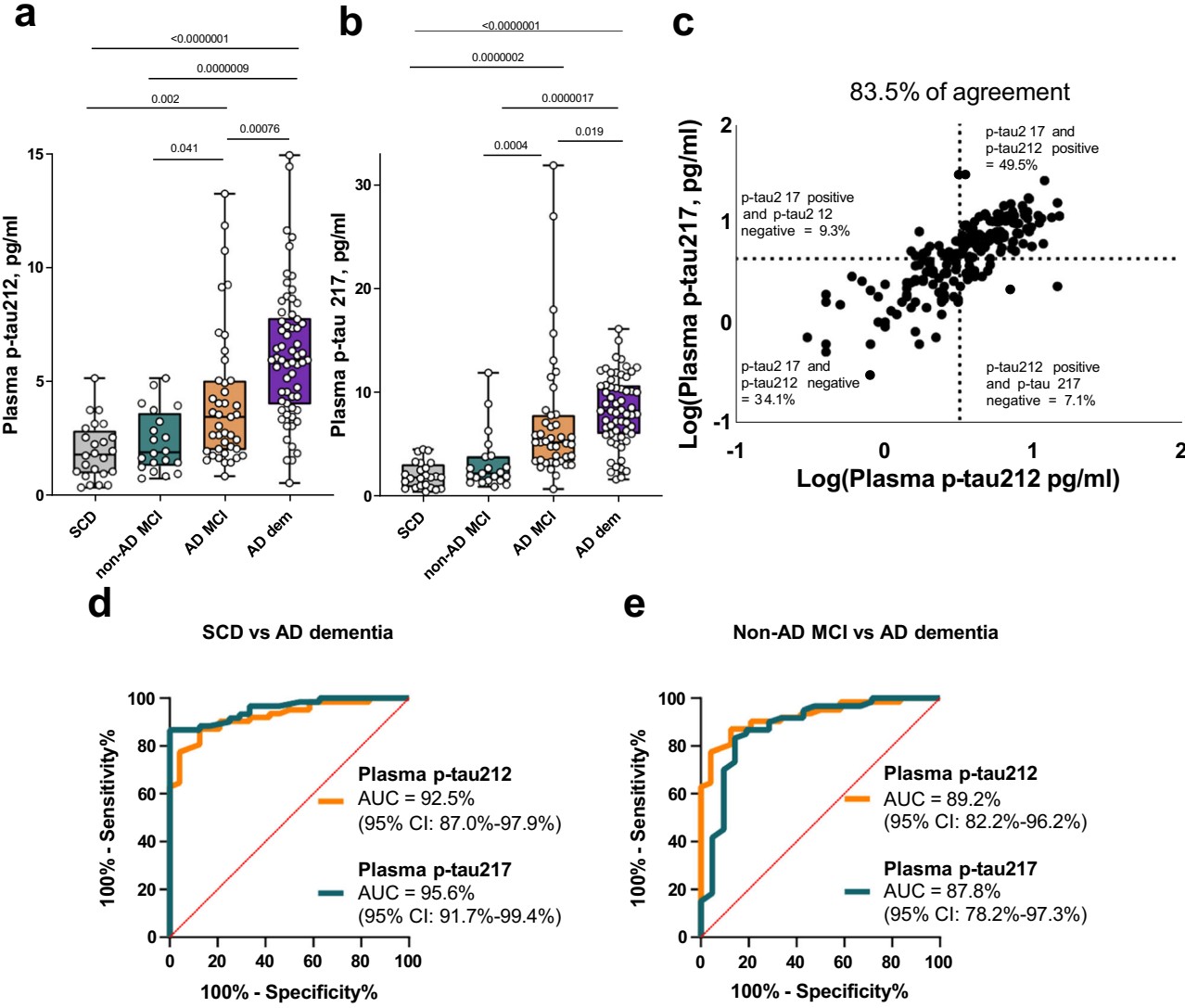

**Fig. 7 | Clinical utility of plasma p-tau212 and p-tau217 in a real-world memory clinic cohort.** Clinical validation of the plasma p-tau212 assay versus p-tau217 in the Slovenia memory clinic cohort. **a** Plasma p-tau212 levels in the different diagnostic groups – SCD (Subjective Cognitive Decline) ($n = 24$), non-AD MCI (non-Alzheimer's Disease Mild Cognitive Impairment) ($n = 20$), AD MCI (Alzheimer's Disease Mild Cognitive Impairment) ($n = 41$), AD dementia (Alzheimer's Disease Dementia) ($n = 62$), and **b** Plasma p-tau217 levels in the different diagnostic groups – SCD ($n = 24$), non-AD MCI ($n = 21$), AD MCI ($n = 41$), AD Dementia ($n = 60$) levels in the different diagnostic groups. The SCD and non-AD MCI groups included Aβ-negative participants while the AD MCI and AD dementia groups were all Aβ-positive. **c** Concordance between plasma p-tau212 and p-tau217. Percentage of concordant measurements are given in the lower left and upper right quadrants whilst the percent of discordant cases are in the lower right and upper left quadrants. Assay cut-offs were estimated using the Youden's index. **d** Area under the curve (AUC) comparison of plasma p-tau212 and plasma p-tau217 to differentiate between SCD and AD-dementia. **e** AUC comparison of plasma p-tau212 and plasma p-tau217 measurements to differentiate between non-AD MCI and AD-dementia participants. De Long's test comparisons of the AUCs did not reveal any significant differences. Box plots are shown as median and interquartile ranges (IQR), boundaries of the whiskers are minimum and maximum values. Group differences were examined using Dwass-Steel-Critchlow-Fligner test. Source data are provided as a Source Data file.

performed using the KingFisher Flex System (Thermo Fisher Scientific). Samples were incubated with the beads for 2 h at room temperature, followed by washes with PBS, PBS 0.05% Triton X-100, PBS, and 50 mM ammonium bicarbonate (AMBIC) and elution with 0.5% formic acid. Quality control samples (which were a pool of several plasma samples) and recombinant tau (0.001 μg per sample) were included in each plate as IP control and to monitor the intensity signal for normalization purposes. Further tau enrichment was performed by adding perchloric acid (15 μl, 60% v/v) to the samples, which induced precipitation of the vast majority of proteins but not tau. After centrifugation at 3,000 $g$ for 30 min at 4 °C, supernatants were transferred to a 96-well SPE plate (Oasis PRiME HLB 96-well μElution plate, 3 mg of sorbent per well; Waters) and desalted. The SPE plate was washed with 2 × 200 μl of 5% methanol (v/v) and eluted into a microtiter plate with

200 μl of 50% acetonitrile and 0.1% trifluoroacetic acid. Then, samples were speed-vac-dried. Tryptic digestion was performed by resuspending the samples with trypsin solution (sequencing grade, Promega) (0.1 μg per sample at a concentration 2.5 μg/ml$^{-1}$ in 50 mM AMBIC) and incubating overnight at 37 °C. After 18 h, proteolysis was quenched with trifluoroacetic acid (TFA) final concentration 0.1%, and samples were lyophilized and stored at −20 °C.

For liquid chromatography–mass spectrometry (LC–MS) analysis, samples were resuspended in 50 μl of 0.01% TFA and run in singlicates. MS analysis was performed on a hybrid Orbitrap mass spectrometer (Lumos, Thermo Fisher Scientific), fitted with an EasySpray nano-ESI ion source. The mass spectrometer was operated in the positive ion mode, with the following settings for the parallel reaction monitoring (PRM) scan: Activation Type: HCD; Detector Type: Orbitrap; Orbitrap

Resolution: 60,000; Scan Range: 250–1200; RF Lens: 30%; Easy-IC: On; Isolation Type: Quadrupole; and Isolation Window: 0.7 *m/z*. Maximum Injection Time, Normalized AGC Target, Optimal Collision Energy and FAIMS Voltage were determined experimentally for each peptide. Heavy-labeled AQUA peptide standards were prepared in a mix with adjusted concentrations for each peptide and spiked in during the sample preparation. LC–MS data were acquired using Xcalibur 4.5 and Tune 3.5 software (Thermo Fisher Scientific) and analyzed with Skyline 22.2 software (McCoss Laboratory, University of Washington). The analysis of the plasma samples was performed blinded to any participant information.

## Immunoprecipitation-mass spectrometry

P-tau212 antibody (4 μg/sample) was conjugated with 1 μg/ul Tosyl-lactivated M-280 Dynabeads™ (ThermoFisher Scientific). Conjugated beads were used to immunoprecipitate 100 fmol/ml of tau protein fragments (ThermoFisher Scientific, Caslo) (Sequeces are enclosed in Supplementary Table 1). Samples were brought to 1 ml total volume of phosphate-buffered saline (0.01 M phosphate buffer, 0.14 M NaCl, pH 7.4; PBS) with a final 0.02% concentration of Triton X-100 and incubated at +4 C overnight. After incubation, the beads and samples were transferred to a KingFisher magnetic particle processor (ThermoFisher Scientific) for automatic washing and further elution of the bound species. 100 μl of 0.5% formic acid (FA) was used as an eluent. Eluates were collected, dried in a vacuum centrifuge and stored at −80 °C until further use. Prior to MS analysis, pellet was trypsin digested using 35 ng of trypsin per sample in 50 mM ammonium bicarbonate. The samples were brought to a final volume of 50 μl with ammonium bicarbonate and shaken overnight at 100 RPM at +37 °C. 2ul of FA were used to stop the reaction. Samples were dried in a vacuum centrifuge and stored at −80 °C until further use.

Nanoflow liquid chromatography (LC)-MS was performed with a Dionex 3000 system coupled to a Q Exactive high resolution hybrid quadrupole–orbitrap mass spectrometer equipped with an electrospray ionization source (both Thermo Fisher Scientific, Inc.) as previously described with minor modifications[41,42]. Briefly, samples immunoprecipitated were reconstituted in 7 μL 8% FA/8% acetonitrile in deionized water, and 6 μL was loaded onto an Acclaim PepMap C18 trap column (length 20 mm, internal diameter 75 μm, particle size 3 μm, pore size 100 Å, Thermo Fisher Scientific) for desalting and sample clean-up. The sample loading buffer was 0.05% TFA/2% acetonitrile in water. Separation was then carried out at a flow rate of 300 nL/min by applying a 50 min long linear gradient from 3% to 40% B using a reversed-phase Acclaim PepMap C18 analytical column (length 150 mm, inner diameter 75 μm, particle size 2 μm, pore size 100 Å, Thermo Fisher Scientific, Inc.) where buffer A was 0.1% formic acid in water and buffer B was 0.1% formic acid/84% acetonitrile in water. The mass spectrometer was operated in data-dependent mode using higher-energy collision-induced dissociation for ion fragmentation. Settings for both MS and MS/MS acquisition were: resolution setting 70 000, 1 microscan, target values $10^6$, trap injection time 250 ms.

## Biomarker measurements

All biomarker assays were performed on the Simoa HD-X platform at the University of Gothenburg. In both plasma and CSF, p-tau181 and p-tau231 were measured using published validated in-house assays[8,10] on the Simoa HD-X platform at the University of Gothenburg. Briefly, p-tau181 antibody (AT270, Invitrogen) was used as a capture antibody and paired n-terminal detection antibody (Tau12; BioLegend). P-tau231 antibody (ADxNeurosciences) was used as a capture and similarly paired with Tau12 antibody. CSF p-tau217 was measured according to the Karikari et. al method[36], and similarly, to previously described immunoassays, p-tau217 antibody (#44-744, Invitrogen) was paired with Tau12 antibody. Plasma p-tau217 was measured using a published and validated in-house assay[31], where sheep monoclonal p-tau217 antibody was

paired with Tau12 antibody. Samples were analyzed in singlicates while internal quality controls (iQC) samples measured at the start and the end of each technical run were used to monitor reproducibility. The CV of these iQC samples, both within- and between-runs, are shown in Supplementary Table 13. For the IP-MS, recovery, and precision experiments, which were performed with a different calibrator lot, the plasma p-tau212 concentrations obtained were adjusted to those in the clinical studies by multiplying by a factor that was determined based on the difference between the same iQC samples that were analyzed with both calibrator lots. Mass spectrometric analysis of plasma samples was performed using and validated method developed at the University of Gothenburg, as previously described[43].

## Gothenburg cohort

The Gothenburg discovery cohort ($n = 30$) included EDTA plasma samples from neurochemically defined Alzheimer's disease participants ($n = 16$) and controls ($n = 14$) selected based on their CSF biomarker profile (CSF Aβ42 < 530 pg/ml, CSF p-tau > 60 pg/ml, and CSF t-tau > 350 pg/ml[8,44]. The Alzheimer's disease group had no evidence of other neurological conditions based on routine clinical and laboratory assessments. The control group consisted of selected patients with a biomarker-negative profile. Demographic characteristics are shown in Supplementary Table 8.

## Polish cohort

The Polish cohort included individuals with paired CSF and EDTA plasma samples ($n = 95$) classified by Erlangen Score (ES)[45] for neurochemically normal (ES = 0) through improbable AD (ES = 1), possible AD (ES = 2 or 3), to probable AD (ES = 4). Participants with ES 1-2 ($n = 21$) were classified as controls and those with ES 3 or higher were classified as AD ($n = 74$). Demographic characteristics are shown in Supplementary Table 9.

## BLSA-Neuropathology cohort

This cohort included participants of the Autopsy Program of the Baltimore Longitudinal Study of Aging (BLSA)[46]. Participants underwent annual evaluations that included multiple neuropsychological tests and clinical examinations that have been widely characterized[47–49]. Post-mortem examination was done at the Division of Neuropathology, Johns Hopkins University, with the details having been described previously[48]. Neuropathological changes were assessed according to The Consortium to Establish a Registry for Alzheimer's Disease (CERAD)[32] or the Braak criteria[50]. Patients were divided into cognitively normal (controls, $n = 12$), cognitively normal within one year before death but have extensive neuropathological changes (ASYMAD, that is people with similar tau and Aβ burden as mild cognitive impairment (MCI) but without cognitive decline; $n = 15$)[34] and AD ($n = 20$)[36,51]. Demographic characteristics are shown in Supplementary Table 6.

## UCSD-Neuropathology cohort

The Neuropathology cohort consisted of CSF samples ($n = 67$) from research participants enrolled in the University of California San Diego (UCSD) Shiley-Marcos Alzheimer's Disease Research Center (ADRC). Annual clinical assessments of participants which included CSF sample collection and consensus clinical diagnoses were described in an earlier article[52]. Standardized protocols were used to perform autopsy[47]. AD pathology was determined using the CERAD[31], Thal[33] Braak[50] and NIA-Reagan criteria[53] to determine Alzheimer's disease neuropathologic changes (ADNC). The "Low pathology" group included Braak 0-II cases who were without Lewy body dementia, hippocampal sclerosis, significant major vascular pathology, imbic-predominant age-related TAR DNA binding protein-43 (TDP-43) encephalopathy, or other neurodegenerative pathology. The Other pathology group included those with brain neurodegenerative changes in the absence of ADNC[52]. Demographic characteristics are shown in Supplementary Table 7.

## Slovenian cohort

This clinical cohort from the Department of Neurology, University Medical Centre Ljubljana, Slovenia, included patients with AD dementia ($n = 62$), non-AD dementia ($n = 39$), mild cognitive impairment (MCI) due to AD (AD MCI; $n = 41$), MCI not due to AD (non-AD MCI; $n = 22$) and participants with subjective cognitive impairment (SCI; $n = 26$). The participants underwent comprehensive clinical examination with neurological and neuropsychological assessment, CSF analysis, and structural and (when deemed necessary) functional neuroimaging. CSF biomarker profile (locally validated cut-offs: $A\beta_{42/40} < 0.077$, p-tau181 > 60 ng/l, tau > 400 ng/L), the dementia DSM V criteria[54] and the Winblad & Peterson MCI diagnostic criteria[55] were used to establish a clinical diagnosis of AD, AD MCI, and non-AD MCI. Patients with MCI and CSF profile of AD continuum (decreased $A\beta_{42/40}$ ratio with or without elevated p-tau181 and tau) were defined as AD MCI, and those with normal AD biomarkers as non-AD MCI group. Considering all clinical and paraclinical findings and the respective diagnostic criteria[56–58]. Individuals with SCI had comparable cognitive performance to the others but with evident self-perceived decline in cognition. Demographic characteristics are shown in Table 1.

## Ethical clearance

This study was performed according to the Declaration of Helsinki. The Gothenburg Discovery cohort that used de-identified leftover clinical samples was approved by the ethics committees at the University of Gothenburg (#EPN140811). Studies, including the Polish cohort were approved by the local ethical committee (KB-380/2017) at the University of Wroclaw. The UCSD-Neuropathology cohort was reviewed and approved by the human subject review board at UCSD. Informed consent was obtained from all patients or their caregivers consistent with California State law (IRB 170957). The study in Slovenian cohort was approved by the Medical Ethics Committee of the Republic of Slovenia, Ministry of Health (0120-342/2021/6). The BLSA studies have ongoing approval from the Institutional Review Board of the National Institute of Environmental Health Science, National Institute of Health (IRB 03AG0325). Anatomical Gift Act for organ donation and a repository consent was signed by the participants to allow sharing the data and biospecimens. We used the term "sex" in the manuscript, based on self-reporting. In post-mortem cohorts "sex" was used as a covariate in statistical analysis. In five analyzed cohorts, samples were coming both from man ($n = 173$) and woman ($n = 215$), therefore we believe that findings presented here apply to both sexes.

## Statistical analyzes

Statistical analyzes were performed with Prism version 9.5 (GraphPad, San Diego, CA, USA) and R Studio version 2023.3.0.386 (RStudio: Integrated Development for R. RStudio, PBC, Boston, MA URL http://www.rstudio.com/). Data are shown as median and interquartile ranges (IQR). The distribution of data sets were examined for normality using the Saphiro-Wilk test. Non-parametric tests were used for non-normally distributed data. We used Receiver Operating Curves and Area Under the Curve (AUC-ROC) to examine diagnostic potential and the DeLong's test (done using R package *pROC*[59]) to compare AUC values for different biomarkers. Spearman correlations were used to establish associations. Fold changes were examined by comparing biomarker values with the mean of the control group. Group differences were examined using two-tailed Mann–Whitney test (two categories) or the Dwass-Steel-Critchlow-Fligner test (PMCMRplus R package) (three or more groups). For the post-mortem cohorts, we used the estimated marginal means (EMMs) (*emmeans* R package) to compare group differences after adjusting for sex, age and CSF/plasma collection-to-death intervals. Analysis of variance (ANOVA) with Tukey's post-hoc test was used to compare differences between groups. Pairwise comparisons were adjusted for multiple comparisons. Significance was set at $p < 0.05$. LC-MS/MS acquisitions were processed using Mascot Daemon v2.6/Mascot Distiller v2.6.3 (both Matrix Science) for charge and isotope deconvolution before submitting searches using Mascot search engine v2.6.1. Searches were made against a custom made tau database; for processing and search settings, refer to the following previous publications[41,42]. Quantitative analysis was performed using Skyline v22.2.0.257 (MacCoss Lab)[60].

## Reporting summary

Further information on research design is available in the Nature Portfolio Reporting Summary linked to this article.

## Data availability

The custom database used for peptide search contained only human tau isoforms from Uniprot (Uniprot ID: P10636). Additional data is provided in the supplementary data. Source data are provided with this file. Any other blinded anonymized data is available on request from the corresponding author. Request will be reviewed by the investigators and respective institutions to verify if the data transfer is in agreement with EU or USA legislation on general data protection or is subject to any intellectual property or confidentiality obligations. Source data are provided with this paper.

## Code availability

The code generated in this study have been deposited in the Zenodo database under accession code: https://zenodo.org/records/10657935.

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

## Acknowledgements

The authors thank all participants of the research cohorts studied here and their caregivers for their invaluable time, support and biospecimen donations. We also thank the research and medical staff at the universities and medical facilities for their collaborative support. Furthermore, we are grateful to Celia Hök Fröhlander for her immense support with biofluid sample processing and aliquoting. PRK was funded by Demensförbundet and Anna Lisa, and Brother Björnsson's Foundation. FG-O was funded by the Anna Lisa and Brother Björnsson's Foundation. LMG is supported by the Brightfocus Foundation (A2022015F), the Swedish Dementia Foundation, Gun and Bertil Stohnes Foundation, Åhlén-stiftelsen, Alzheimerfonden (AF-968621), and Gamla Tjänarinnor Foundation. HZ is a Wallenberg Scholar supported by grants from the Swedish Research Council (#2022-01018 and #2019-02397), the European Union's Horizon Europe research and innovation programme under grant agreement No 101053962, Swedish State Support for Clinical Research (#ALFGBG-71320), the Alzheimer Drug Discovery Foundation (ADDF), USA (#201809-2016862), the AD Strategic Fund and the Alzheimer's Association (#ADSF-21-831376-C, #ADSF-21-831381-C, and #ADSF-21-831377-C), the Bluefield Project, the Olav Thon Foundation, the Erling-Persson Family Foundation, Stiftelsen för Gamla Tjänarinnor, Hjärnfonden, Sweden (#FO2022-0270), the European Union's Horizon 2020 research and innovation programme under the Marie Skłodowska-Curie grant agreement No 860197 (MIRIADE), the European Union Joint Programme – Neurodegenerative Disease Research (JPND2021-00694), the National Institute for Health and Care Research University College London Hospitals Biomedical Research Centre, and the UK Dementia Research Institute at UCL (UKDRI-1003). KB is supported by KB is supported by the Swedish Research Council (#2017-00915), the Alzheimer Drug Discovery Foundation (ADDF), USA (#RDAPB-201809-2016615), the Swedish Alzheimer Foundation (#AF-930351, #AF-939721 and #AF-968270), Hjärnfonden, Sweden (#FO2017-0243 and #ALZ2022-0006), the Swedish state under the agreement between the Swedish government and the County Councils, the ALF-agreement (#ALFGBG-715986 and #ALFGBG-965240), the European Union Joint Program for Neurodegenerative Disorders (JPND2019-466-236), the National Institute of Health (NIH), USA, (grant #1R01 AG068398-01), and the Alzheimer's Association 2021 Zenith Award (ZEN-21-848495). TKK was supported by the NIH (R01 AG083874-01, U24 AG082930-01, 1 RF1 AG052525-01A1, 5 P30 AG066468-04, 5 R01 AG053952-05, 3 R01 MH121619-04S1, 5 R37 AG023651-18, 2 RF1 AG025516-12A1, 5 R01 AG073267-02, 2 R01 MH108509-06, 5 R01 AG075336-02, 5 R01 AG072641-02, 2 P01 AG025204-16), the Swedish Research Council (Vetenskåpradet; #2021-03244), the Alzheimer's Association (#AARF-21-850325), the Swedish Alzheimer Foundation (Alzheimerfonden), the Aina (Ann) Wallströms and Mary-Ann Sjöbloms stiftelsen, and the Emil och Wera Cornells stiftelsen.

## Author contributions

P.R.K., M.D., S.K., A.E., Y.A., Performed statistical analysis; P.R.K., K.B., T.K.K., designed and developed the new p-tau212 simoa assay; P.R.K., F.G.O., N.J.A., H.K, H.Z., K.B., T.K.K., performed simoa experiments, analyzed the result, and were responsible for the discovery cohort management. Y.A., V.R.V., A.M., J.C.T., S.M.R., and M.Th. were responsible for the BLSA post-mortem cohort management; D.S., D.G. were responsible for the UCSD Neuropathology cohort management; M.D., A.K.P., I.W., B.P., B.M., and P.L. were responsible for the Polish cohort management; A.E., S.C., M.G.K., and U.R. were responsible for the Slovenian Cohort Management; T.L. performed immunohistochemistry stainings; S.K., J.H. Performed fluorescent stainings; M.Tu. and P.H. designed and validated the p-tau212 and p-tau217 antibodies using ELISA. E.C. and G.B. IP-Mass spectrometry analysis; L.M.G. Plasma Mass spectrometry analysis; P.K.K., T.K.K., and K.B. Wrote the manuscript. All authors edited and revised the manuscript.

## Funding

## Competing interests

M.T. and P.H. are employees of Bioventix Plc. H.Z. has served at scientific advisory boards and/or as a consultant for Abbvie, Acumen, Alector, Alzinova, ALZPath, Annexon, Apellis, Artery Therapeutics, AZTherapies, CogRx, Denali, Eisai, Nervgen, Novo Nordisk, Optoceutics, Passage Bio, Pinteon Therapeutics, Prothena, Red Abbey Labs, reMYND, Roche, Samumed, Siemens Healthineers, Triplet Therapeutics, and Wave, has given lectures in symposia sponsored by Cellectricon, Fujirebio, Alzecure, Biogen, and Roche. K.B. has served as a consultant or at advisory boards for Abcam, Axon, BioArctic, Biogen, JOMDD/Shimadzu. Julius Clinical, Lilly, MagQu, Novartis, Ono Pharma, Pharmatrophix, Prothena, Roche Diagnostics, and Siemens Healthineers. H.Z. and K.B. are co-founders of Brain Biomarker Solutions in Gothenburg AB, a GU Ventures-based platform company at the University of Gothenburg. NJA has given lectures in symposia sponsored by Lilly, BioArctic, and Quanterix. The other authors declare no competing interests.

## Additional information

[1]Department of Psychiatry and Neurochemistry, Institute of Neuroscience and Physiology, The Sahlgrenska Academy at the University of Gothenburg, Mölndal 431 80, Sweden. [2]Clinical Neurochemistry Laboratory, Sahlgrenska University Hospital, Mölndal 431 80, Sweden. [3]Department of Neurology, University Medical Centre Ljubljana, Ljubljana 1000, Slovenia. [4]Faculty of Pharmacy, University of Ljubljana, Ljubljana, Slovenia. [5]Bioventix Plc, Farnham GU9 7SX, UK. [6]Brain Aging and Behavior Section, Laboratory of Behavioral Neuroscience, National Institute on Aging, National Institutes of Health, Baltimore, MD 21224, USA. [7]Department of Neurosciences, University of California, San Diego, CA 92161, USA. [8]Department of Neurodegeneration Diagnostics, Medical University of Białystok, Białystok 15-269, Poland. [9]Clinical and Translational Neuroscience Section, Laboratory of Behavioral Neuroscience, National Institute on Aging, National Institutes of Health, Baltimore, MD 21224, USA. [10]Department of Old Age Psychiatry, King's College London, London SE5 8AF, UK. [11]Centre for Age-Related Medicine, Stavanger University Hospital, 4011 Stavanger, Norway. [12]South London & Maudsley NHS Foundation, NIHR Biomedical Research Centre for Mental Health & Biomedical Research Unit for Dementia, SE5 8AF, London, UK. [13]Dementia Disorders Center, Medical University of Wrocław, 59-330 Ścinawa, Poland. [14]Department of Neurology, Medical University of Wrocław, 50-556 Wrocław, Poland. [15]Department of Neurology, Johns Hopkins University School of Medicine, Baltimore, MD 21287, USA. [16]Department of Pathology, John Hopkins University School of Medicine, Baltimore, MD 21287, USA. [17]Department of Neurodegenerative diseases, UCL Queen Square Institute of Neurology, WC1N 1PJ London, UK. [18]Laboratory of Behavioral Neuroscience, National Institute on Aging, National Institutes of Health, Baltimore, MD 21224, USA. [19]Faculty of Medicine, University of Ljubljana, Ljubljana, Slovenia. [20]Karolinska Institutet, Department of Neurobiology, Care Sciences and Society, Division of Clinical Geriatrics, 141 52 Huddinge, Sweden. [21]Department of Neurodegenerative Disease, Dementia Research Centre, UCL Institute of Neurology, Queen Square, London WC1E 6BT, UK. [22]Department of Rheumatology, University Medical Center Ljubljana, Ljubljana, Slovenia. [23]UK Dementia Research Institute, University College London, London WC1E 6BT, UK. [24]Hong Kong Center for Neurodegenerative Diseases, HKCeND, Hong Kong 1512-1518, China. [25]School of Medicine and Public Health, University of Wisconsin-Madison, Madison, WI 53726, USA. [26]Department of Psychiatry and Psychotherapy, Universitätsklinikum Erlangen, and Friedrich-Alexander Universität Erlangen-Nürnberg, Erlangen 91054, Germany. [27]Department of Psychiatry, School of Medicine, University of Pittsburgh, Pittsburgh, PA 15213, USA. [28]These authors jointly supervised this work: Kaj Blennow, Thomas K. Karikari. ✉e-mail: przemyslaw.kac@gu.se

