## [Peer Review File · Nature Communications]

Plasma p-tau₂₁₂ antemortem diagnostic performance and prediction of autopsy verification of Alzheimer's disease neuropathologyREVIEWER COMMENTS

Reviewer #1 (Remarks to the Author):

Kac et al. Report about a novel antibody that detects plasma p-tau212. The authors describe the biochemical characterization of the antibody, the validity testing by performing immunohistochemistry and a comparison with clinical standard measures for AD. In addition, they used two cohort of 47 and 67 cases with neuropathological post-mortem status. The blood p-tau212 Simoa showed a similar detection of p-tau as the plasma p-tau217 Simoa. The authors could also demonstrate that their new marker distinguished convincingly between Braak stage I/II, III/IV and V/VI cases as well as between CERAD sparse, moderate and frequent neuritic plaques, and between control, asymptomatic and symptomatic AD cases. The authors conclude that p-tau212 is a novel peripheral biomarker that provides insights into AD pathophysiology.

The results presented by the authors appear to be quite convincing and introduce a novel AD biomarker candidate. This could have impact for the clinical diagnosis of AD as this biomarkers appears to be quite sensitive and reliable.

However, there are a few points that should be addressed before this manuscript is ready for publication:

1. For the antibody quantification, I would expect western blots documenting the specific staining of p-tau212 without interacting with other tau epitopes. For this 1. blots with synthetic p-tau212, p-tau181, p-tau231, p-tau202/205 need to be shown and 2. western blots from brain homogenates and blood to clarify that the antibodies do not cross react with other proteins.
2. The authors provide no correlation with amyloid plaque pathology in their neuropathological cohorts (Thal amyloid phase). Since it has been shown that plasma p-tau usually correlates better with amyloid than with tau pathology, this correlation needs to be shown as well in the neuropathological cohorts.
3. A major limitation of this manuscript is that no separate tau pathology negative control group (=Braak stage 0, Thal amyloid phase 0) was included in the neuropathological cohorts. Accordingly, the authors cannot be sure that even in the Braak stage I/II cases the tau levels are already increased although still being very low compared to the symptomatic AD cases. This need to be mentioned in the "Limitations" section of the Discussion.
4. The stainings show for immunohistochemistry are restricted to very few high-power images. This is not very convincing. I would expect that the authors depict low power pictures as well that document a similar regional distribution of p-tau212 as seen with AT8. The medial temporal lobe would be perfect to convince the reader that p-tau212 antibodies stain similar as AT8.

Reviewer #2 (Remarks to the Author):

The well-written and clear manuscript titled "Plasma p-tau212: antemortem diagnostic performance and prediction of autopsy verification of Alzheimer's disease neuropathology" is an important contribution to the ongoing research on Alzheimer's disease blood-based biomarkers. I enjoyed reading it. The methods employed to develop the antibodies and assays appear robust, the clinical validation performed was extensive by measuring 5 independent cohorts (total n=388). I have some suggestions for the authors to consider:

- Authors titrated the antibody concentrations (coated on ELISA plates) against one concentration of biotin-coated antigen to show their antibodies are highly selective for either P-tau212 or P-tau217. They additionally conducted experiments including an antigen phosphorylated both at 212 and 217, and lastly an antigen phosphorylated at 212, the neighboring 214 and 217 to further show selectivity of their antibodies. These experiments are well-conducted and clear. If possible, the figure 1 and manuscript would benefit from including an experiment for titrated P-tau212 and P-tau217 antibody concentrations and their selectivity for an biotin-coupled antigen phosphorylated at only 214. Additionally, authors used their monoclonal antibodies to develop clinical immunoassays. I suggest authors to conduct selectivity/specificity experiments in these assay format as well. With the ultimate assay reagents and parameters as finetuned and

technically validated by the authors, titrate different amounts of each antigen in buffer (e.g. blank, low, medium, high x p-tau212, p-tau217, p-tau214, p-tau181, p-tau231, and perhaps the combinations as also used in figure 1) and test signals for both assays. Also, incubate combinations of P-tau antigens together in different combinations and concentrations to mimic physiological samples, were 181/231/212/217 are present together: test if presence of other antigens interfere with the immunoassay signal / antibody-antigen-antibody complexes formation of both assays.

- Paragraph 'Development and validation of blood-based assays for p-tau212 and p-tau217': this paragraph describes the P-tau212 assays for CSF and plasma, the results of the novel P-tau217 assay is lacking. Could authors add those results, and update the paragraph title to match the content? Also, it would be helpful to read the %parallelism values for each subsequent dilution factor (i.e. 1 to 2-fold dilution, 2- to 4-fold dilution etc). How was LLOQ determined? Was this based on e.g. blank readings, or %CV plots of duplicate measurements of the clinical cohorts? Extended figure 1: please add P-tau217 validation results. Many readers are used to read intra- and inter-assay reproducibility as %CVs, could authors add those numbers also? Extended table 1: title suggest P-tau217 results are there but those are lacking, please add. Was the recovery experiment only conducted once, for one spike concentration, or are these values averages? It is common to validate spike recovery with three concentrations (low, medium, high as compared to the standard curve and physiological concentrations in samples), in at least three individual samples to obtain averages.

- Authors present number of samples measured above the LLOQ. I suggest authors to also add %CV plots for the duplicate measurements (%CV on y-axis, concentration at x-axis) for all assays. Those additionally show robustness of the assays to measure the samples accurately, and give extra information on top of %samples that are measured above the LLOQ threshold.

- Do I understand correctly from figure 3 that the BLSA cohort was a plasma with autopsy cohort, and the UCSD was a CSF with autopsy cohort? this could be mentioned in the 'participants' section, to make the cohort descriptions more clear and the results easier to interpret. In the legend, P-tau217 is written where it should be P-tau212 (authors are advised to carefully check this throughout the manuscript and tables/figures). Regarding the P-tau181, P-tau231, P-tau212 and P-tau217 comparison in extended table 2, I would appreciate to also see those findings in boxplots, to see the variation in the measurements, which is not captured by a fold-change only. Also, in my opinion this might be part of the main body of the text as for validation it is important to compare the new P-tau biomarker to the current P-tau biomarkers.

- Authors conclude that plasma and CSF P-tau212 have comparable performance, which was seemingly based on the small cohort of Poland, where the AUCs for plasma and CSF were not significantly different though they were numerically different, and also the fold-change of CSF was much larger compared to plasma. I am not sure this conclusion is fully supported, and might need verification in a larger n.

- Gothenburg cohort: this is placed in the plasma versus CSF P-tau212 paragraph, which is not subject of the Gothenburg comparisons. Also, authors did not draw a conclusion on these results while they did on the plasma versus CSF P-tau212 comparison for P-tau212. I suggest authors to carefully check all results headings in light of the presented results in that paragraph.

- Authors developed in-house P-tau217 assays, but in methods 'Biomarker measurements' I read that P-tau217 CSF was measured with the Karikari assay. The earlier paragraph suggests the novel P-tau217 assay should be used as it does not have cross-reactivity with P-tau212 as opposed to some other P-tau217 assays, and results should be more similar to P-tau212 results since assay conditions and reagents were kept the same as much as possible. Why was this novel P-tau217 assay not used for CSF in the clinical validation comparisons? Does the Karikari assay show cross-reactivity with P-tau212?

- Did authors consider to also include P-tau181 and P-tau231 in all the assessed cohorts to include in the diagnostic value comparisons?

- The P-tau212 and P-tau217 show a very high degree of similarity in diagnostic performance, but also in binding in the histology experiments. Is there added value to have both markers in this cohort? Could authors also discuss the potential role they foresee for P-tau212 in diagnosis and management of AD?

- Throughout the paper, it seems that P-tau212 and P-tau217 have very similar performance, which is larger than for P-tau231 and P-tau181. Could authors discuss if this could be technically or biologically driven?

- Should authors include a 'reference P-tau217 assay' in their comparisons, to validate their novel

assay against well-established and well-validated assays? For example the Gothenbrug mass spect assay?

- Could authors reflect on future validations required for the novel P-tau212 biomarker?

Reviewer #1

Kac et al. Report about a novel antibody that detects plasma p-tau212. The authors describe the biochemical characterization of the antibody, the validity testing by performing immunohistochemistry and a comparison with clinical standard measures for AD. In addition, they used two cohort of 47 and 67 cases with neuropathological post-mortem status. The blood p-tau212 Simoa showed a similar detection of p-tau as the plasma p-tau217 Simoa. The authors could also demonstrate that their new marker distinguished convincingly between Braak stage I/II, III/IV and V/VI cases as well as between CERAD sparse, moderate and frequent neuritic plaques, and between control, asymptomatic and symptomatic AD cases.

The authors conclude that p-tau212 is a novel peripheral biomarker that provides insights into AD pathophysiology. The results presented by the authors appear to be quite convincing and introduce a novel AD biomarker candidate. This could have impact for the clinical diagnosis of AD as this biomarker appears to be quite sensitive and reliable.

We thank the reviewer for their succinct summary of our work.

However, there are a few points that should be addressed before this manuscript is ready for publication:

1. For the antibody quantification, I would expect western blots documenting the specific staining of p-tau212 without interacting with other tau epitopes. For this
 1. blots with synthetic p-tau212, p-tau181, p-tau231, p-tau202/205 need to be shown and
 2. western blots from brain homogenates and blood to clarify that the antibodies do not cross react with other proteins.

1. *Instead of western blots which have limited sensitivity, we have performed immunoprecipitation-mass spectrometry experiments to confirm that the p-tau212 antibody does not interact with neighbouring p-tau epitopes. We have included these results as supplementary data (Supplementary Data Table Figure 1).*
 2. *We have performed preliminary immuno-precipitation-mass spectrometry (IP-MS) experiments in TBS-soluble brain fraction samples, to verify that our antibody has no other identified targets in the brain, and it confirmed specificity of the antibody (n=1; data not included in the publication).*

2. The authors provide no correlation with amyloid plaque pathology in their neuropathological cohorts (Thal amyloid phase). Since it has been shown that plasma p-tau usually correlates better with amyloid than with tau pathology, this correlation needs to be shown as well in the neuropathological cohorts.

We have now added correlations of plasma p-tau with Thal phase for the UCSD-Neuropathology Cohort (Page 9, Figure 3g; Extended Data Table 2; Supplementary Data Figure 6), Additionally, we performed statistical analysis and added box-plots for all measured p-tau biomarkers. (Supplementary Data Figure 6)

3. A major limitation of this manuscript is that no separate tau pathology negative control group (=Braak stage 0, Thal amyloid phase 0) was included in the neuropathological cohorts. Accordingly, the authors cannot be sure that even in the Braak stage I/II cases the tau levels are already increased although still being very low compared to the symptomatic AD cases. This need to be mentioned in the "Limitations" section of the Discussion.

We agree with the reviewer, and we included this in the limitation section of the discussion. "Limitations of this study include lack of separate Braak and Thal phase 0 negative control group." (Page 14)

4. The stainings show for immunohistochemistry are restricted to very few high-power images. This is not very convincing. I would expect that the authors depict low power pictures as well that document a similar regional distribution of p-tau212 as seen with AT8. The medial temporal lobe would be perfect to convince the reader that p-tau212 antibodies stain similar as AT8.

We were unable to perform immunohistochemical stainings due to lack of required brain tissue.

Reviewer #2 (Remarks to the Author):

The well-written and clear manuscript titled "Plasma p-tau212: antemortem diagnostic performance and prediction of autopsy verification of Alzheimer's disease neuropathology" is an important contribution to the ongoing research on Alzheimer's disease blood-based biomarkers. I enjoyed reading it. The methods employed to develop the antibodies and assays appear robust, the clinical validation performed was extensive by measuring 5 independent cohorts (total n=388).

Thank you for your kind comments.

I have some suggestions for the authors to consider:

- Authors titrated the antibody concentrations (coated on ELISA plates) against one concentration of biotin-coated antigen to show their antibodies are highly selective for either P-tau212 or P-tau217. They additionally conducted experiments including an antigen phosphorylated both at 212 and 217, and lastly an antigen phosphorylated at 212, the neighboring 214 and 217 to further show selectivity of their antibodies. These experiments are well-conducted and clear.

If possible, the figure 1 and manuscript would benefit from including **an experiment for titrated P-tau212 and P-tau217 antibody concentrations and their selectivity for an biotin-coupled antigen phosphorylated at only 214.**

We have now validated selectivity of our p-tau212 antibodies against biotin-coupled peptide phosphorylated at serine-214. The experiment results are now added to Figure 1 (See Figure 1g)

Additionally, authors used their monoclonal antibodies to develop clinical immunoassays. I suggest authors to conduct selectivity/specificity experiments in these assay format as well. With the ultimate assay reagents and parameters as finetuned and technically validated by the authors, titrate different amounts of each antigen in buffer (e.g. blank, low, medium, high x p-tau212, p-tau217, p-tau214, p-tau181, p-tau231, and perhaps the combinations as also used in figure 1) and test signals for both assays.

We encountered multiple obstacles with constructing or ordering a peptide, that is only phosphorylated at the desired specific epitope, and concurrently has the N-terminal tau region that is targeted by the detection antibody in the Simoa assays. Therefore, we were unable to perform the suggested experiments. However, we have added new plots that demonstrate lack of binding of the p-tau212 antibody to p-tau181 or p-tau231 epitopes to Figure 1 (see Figure 1h-i).

Also, incubate combinations of P-tau antigens together in different combinations and concentrations to mimic physiological samples, were 181/231/212/217 are present together: test if presence of other antigens interfere with the immunoassay signal / antibody-antigen-antibody complexes formation of both assays.

We have performed IP-MS experiments using a mix of different p-tau peptides to verify if the presence of other peptides (other than the p-tau212 site) interfere with the binding of the p-tau212 antibody to its epitope. No other peptides aside those which were phosphorylated at T212 or T212+T217 were captured (Supplementary Data Table 1).

- Paragraph 'Development and validation of blood-based assays for p-tau212 and p-tau217': this paragraph describes the P-tau212 assays for CSF and plasma, the results of the novel P-tau217 assay is lacking. Could authors add those results, and update the paragraph title to match the content?

We have updated the paragraph and added the reference to a recent publication (PMID: 37975513) that describes the new p-tau217 assay including its validation data.

Also, it would be helpful to read the %parallelism values for each subsequent dilution factor (i.e. 1 to 2-fold dilution, 2- to 4-fold dilution etc).

We have added %Parallelism values as a supplementary table (Supplementary Data Table 3).

How was LLOQ determined? Was this based on e.g. blank readings, or %CV plots of duplicate measurements of the clinical cohorts?

The LLOQ was determined based on the calibrator points, at which relevant discrepancy between %CV of the signal and of the sample concentration was observed.

Extended figure 1: please add P-tau217 validation results. Many readers are used to read intra- and inter-assay reproducibility as %CVs, could authors add those numbers also?

P-tau217 validation results are presented in a newly published article (PMID: 37975513). Following the reviewer's suggestion, we have added %CV data.

Extended table 1: title suggest P-tau217 results are there but those are lacking, please add. Was the recovery experiment only conducted once, for one spike concentration, or are these values averages? It is common to validate spike recovery with three concentrations (low, medium, high as compared to the standard curve and physiological concentrations in samples), in at least three individual samples to obtain averages.

P-tau217 recovery experiments were performed, and the results included in the said publication (PMID: 37975513).

For the p-tau212 recovery, the experiment was conducted for one sample using a concentration of the spike analyte. Following the reviewer suggestion, we have now performed the experiment one more time in two separate samples for both plasma and CSF using two different concentrations of the spike material (high and low).

- Authors present number of samples measured above the LLOQ. I suggest authors to also add %CV plots for the duplicate measurements (%CV on y-axis, concentration at x-axis) for all assays. Those additionally show robustness of the assays to measure the samples accurately, and give extra information on top of %samples that are measured above the LLOQ threshold.

For the clinical cohorts, we measured the samples in singlicates, due to limited volumes. However, we used internal quality controls at the beginning and at the end of the run, and in duplicates. These values are presented in Supplementary Data Table 4.

Nonetheless, we have added a %CV plot for the plasma samples ran in duplicates, to the supplementary data (Supplementary Data Figure 4).

- Do I understand correctly from figure 3 that the BLSA cohort was a plasma with autopsy cohort, and the UCSD was a CSF with autopsy cohort? this could be mentioned in the 'participants' section, to make the cohort descriptions more clear and the results easier to interpret.

Yes, BLSA had paired antemortem plasma with autopsy information whilst UCSD had matched antemortem CSF with autopsy data. We have now clarified this information in the relevant section of the Results and Methods as well the legends of the affected figures.

In the legend, P-tau217 is written where it should be P-tau212 (authors are advised to carefully check this throughout the manuscript and tables/figures).

We regret this oversight, which has now been rectified. We thank the reviewer for bringing this to our attention.

Regarding the P-tau181, P-tau231, P-tau212 and P-tau217 comparison in extended table 2, I would appreciate to also see those findings in boxplots, to see the variation in the measurements, which is not captured by a fold-change only. Also, in my opinion this might be part of the main body of the text as for validation it is important to compare the new P-tau biomarker to the current P-tau biomarkers.

Following the reviewer's suggestion, we have added boxplots for the different biomarkers as a supplementary figure (Supplementary Data Figure 6).

- Authors conclude that plasma and CSF P-tau212 have comparable performance, which was seemingly based on the small cohort of Poland, where the AUCs for plasma and CSF were not significantly different though they were numerically different, and also the fold-change of CSF was much larger compared to plasma. I am not sure this conclusion is fully supported, and might need verification in a larger n.

We agree with the reviewer, and we have indicated in the discussion section that the use of a relatively small cohort to evaluate the comparability of the performances of plasma and CSF p-tau212 is a limitation point of this research. We have included this limitation in the manuscript "Additionally, the Polish cohort, which was used for the comparison of p-tau212 immunoassay performance in plasma and CSF consisted of n=95 participants, and verification of the assay performance in larger cohorts would be favourable." (Page 14)

- Gothenburg cohort: this is placed in the plasma versus CSF P-tau212 paragraph, which is not subject of the Gothenburg comparisons. Also, authors did not draw a conclusion on these results while they did on the plasma versus CSF P-tau212 comparison for P-tau212. I suggest authors to carefully check all results headings in light of the presented results in that paragraph.

We are thankful for this comment. The heading was referring to the Polish Cohort. We cross checked this and corrected the headings in the light of the presented results.

- Authors developed in-house P-tau217 assays, but in methods 'Biomarker measurements' I read that P-tau217 CSF was measured with the Karikari assay. The earlier paragraph suggests the novel P-tau217 assay should be used as it does not have cross-reactivity with P-tau212 as opposed to some other P-tau217 assays, and results should be more similar to P-tau212 results since assay conditions and reagents were kept the same as much as possible. Why was this novel P-tau217 assay not used for CSF in the clinical validation comparisons? Does the Karikari assay show cross-reactivity with P-tau212?

For the UCSD cohort, the earlier CSF p-tau217 assay had been measured before the new assays were developed. Moreover, the old CSF p-tau217 assay ("the Karikari method") showed selectivity/specificity to p-tau217. In immunoprecipitation-mass spectrometry experiments using human CSF samples from a memory clinic, the anti-rabbit anti-p-tau217 antibody used in that assay retrieved tau forms that were exclusively phosphorylated at threonine-217, meaning it was specific to the indicated phospho-site. These information were presented in the primary paper for that assay (Karikari et al., 2021 Alzheimer's and Dementia).

In the current study, we decided to switch over to generating a new p-tau217 antibody using an identical platform as what we used for the p-tau212 antibody – sheep monoclonals employing an identical antibody production technology from the same company.

- Did authors consider to also include P-tau181 and P-tau231 in all the assessed cohorts to include in the diagnostic value comparisons?

We included p-tau181 and p-tau231 comparisons in the larger cohorts for which there was sufficient sample volume to examine these markers. In the earlier cohorts examined, however, limited sample volume meant that we had to prioritize showing initial validation of p-tau212 and when possible, its comparison to p-tau217.

- The P-tau212 and P-tau217 show a very high degree of similarity in diagnostic performance, but also in binding in the histology experiments. Is there added value to have both markers in this cohort? Could authors also discuss the potential role they foresee for P-tau212 in diagnosis and management of AD?

In the comparison of p-tau212 and p-tau217 in memory clinic cohort, we found that there were participants who were below the Youden's index driven estimated cut-offs for either p-tau212 or p-tau217. The biomarkers appear to be very similar, however there could be distinct molecular mechanisms underlying the pathology, and this might reflect on clinical diagnosis. Yet, to fully confirm p-tau212 utility in the diagnosis and management of AD, we need to measure levels of this biomarker in other cohorts, for example pre-clinical AD cohort, longitudinal cohort, and imaging cohorts.

- Throughout the paper, it seems that P-tau212 and P-tau217 have very similar performance, which is larger than for P-tau231 and P-tau181. Could authors discuss if this could be technically or biologically driven?

P-tau212 is a target for multiple kinases and phosphatases, and proportion of the phosphorylation and dephosphorylation is different for this epitope. There are multiple molecular processes involved in AD pathology, and our bibliographically driven approach strongly suggest that higher performances of p-tau212 and p-tau217 are results of the development of the disease, rather than technicalities of the assay.

- Should authors include a 'reference P-tau217 assay' in their comparisons, to validate their novel assay against well-established and well-validated assays? For example the Gothenbrug mass spect assay?

Following the reviewer suggestion, we have added comparison of the p-tau212 assay with the Gothenburg mass spectrometry assay which was published recently (PMID: 37198279).

- Could authors reflect on future validations required for the novel P-tau212 biomarker?

We included the reflection in the discussion section at page 15.

"Additionally, the close proximity of p-tau212 to p-tau217 makes it very promising target that could be longitudinally associated with disease progression and conversion from cognitively unimpaired (CU) people to dementia. Moreover, p-tau212 needs to be validated against low-threshold centiloid positivity in CU cohort, to verify its utility in the pre-symptomatic phases. Furthermore, p-tau212 levels were proven to be increased in AD-DS brains, therefore this biomarker might find utility as a biomarker for this population. "

REVIEWERS' COMMENTS

Reviewer #1 (Remarks to the Author):

This is the revised version of a previously submitted manuscript. The authors addressed most of my points sufficiently.

The only point that requires further revision is the quality of the immunohistochemistry figures that are not convincing. To be more convincing low power figures with zoomed in high power figures of the neurofibrillary tangles exhibiting a fibrillar texture would be essential to demonstrate the distribution of the changes in the cortex section.

In the rebuttal that authors state that they were unable to perform immunohistochemical stainings due to a lack of brain tissue. In my opinion, this argument is not convincing. The authors should still have the sections used for figure 2. These sections could be reviewed and used for taking new low power and zoomed in high power images. In my experience, immunofluorescence signals can still be reobserved after a first analysis.

Moreover, one of the authors is affiliated with the Queens Square Neurology Institute which hosts one of the best brain banks for neurodegenerative diseases worldwide. Therefore, lack of tissue does not convince me, and I would strongly encourage the authors to provide better images.

Reviewer #2 (Remarks to the Author):

Thanks to the author for considering my comments. No more questions. Congratulations with the great work!

REVIEWERS' COMMENTS

Reviewer #1 (Remarks to the Author):

This is the revised version of a previously submitted manuscript. The authors addressed most of my points sufficiently.

The only point that requires further revision is the quality of the immunohistochemistry figures that are not convincing. To be more convincing low power figures with zoomed in high power figures of the neurofibrillary tangles exhibiting a fibrillar texture would be essential to demonstrate the distribution of the changes in the cortex section.

In the rebuttal that authors state that they were unable to perform immunohistochemical stainings due to a lack of brain tissue. In my opinion, this argument is not convincing. The authors should still have the sections used for figure 2. These sections could be reviewed and used for taking new low power and zoomed in high power images. In my experience, immunofluorescence signals can still be reobserved after a first analysis.

Moreover, one of the authors is affiliated with the Queens Square Neurology Institute which hosts one of the best brain banks for neurodegenerative diseases worldwide. Therefore, lack of tissue does not convince me, and I would strongly encourage the authors to provide better images.

We have performed immunohistochemistry for p-tau212, and zoomed in figures, where all the major tau pathologies are stained - tangles, neurites, and threads (Fig.2, main manuscript file).

Reviewer #2 (Remarks to the Author):

Thanks to the author for considering my comments. No more questions. Congratulations with the great work!

We are very thankful for the appreciation.